# Whole genome sequencing analysis of body mass index identifies novel African ancestry-specific risk allele

Obesity is a major public health crisis associated with high mortality rates. Previous genome-wide association studies (GWAS) investigating body mass index (BMI) have largely relied on imputed data from European individuals. This study leveraged whole-genome sequencing (WGS) data from 88,873 participants from the Trans-Omics for Precision Medicine (TOPMed) Program, of which 51% were of non-European population groups. We discovered 18 BMI-associated signals ($P < 5 \times 10^{-9}$), including two secondary signals. Notably, we identified and replicated a novel low-frequency single nucleotide polymorphism (SNP) in *MTMR3* that was common in individuals of African descent. Using a diverse study population, we further identified two novel secondary signals in known BMI loci and pinpointed two likely causal variants in the *POC5* and *DMD* loci. Our work demonstrates the benefits of combining WGS and diverse cohorts in expanding current catalog of variants and genes confer risk for obesity, bringing us one step closer to personalized medicine.

In 2015, approximately 12% of adults worldwide had obesity[1], and four years later, the global obesity-related deaths amounted to five million, translating to an age-standardized mortality rate of 62.6 per 100,000 individuals in 2019[2]. Previous genome-wide association studies (GWAS) have identified hundreds of loci associated with obesity-related traits, primarily with body mass index (BMI) – a practical and widely used proxy of overall adiposity. However, most of these genome-wide screens relied on meta-analyses of imputed data, predominantly from individuals of European ancestry[3,4].

Despite making some advancements toward improving ancestral diversity in GWAS, genetic ancestry-stratified analyses and multi-ancestry analyses leveraged for discovery and fine-mapping are uncommon and largely underpowered by comparison. Furthermore, follow-up investigations for known BMI loci identified in European-ancestry populations are insufficiently conducted to evaluate the generalizability of these loci. As such, the majority of BMI-risk variants are common variants (minor allele frequency [MAF] > 5%) in primarily European-ancestry populations, most of which exhibit small effect sizes. While these index variants collectively explain less than 5% of the total phenotypic variation in BMI[5], it is estimated that as much as 1/5 of

the phenotypic variance can be captured by common variants across the entire genome[5], leaving low-frequency and rare variants (MAF ≤ 5%) with potentially large effects to be explored[6].

Whole-genome sequencing (WGS) outperforms genotyping arrays in capturing low and rare frequency variants, as demonstrated in a recent study where researchers revealed that the heritability of BMI estimated using WGS data was comparable to the pedigree-based estimates, $h^2 \approx 0.40$[7]. Thus, the discrepancy between phenotypic variance explained by genetic variations in GWAS compared to the expected heritability may be due to the use of imputed genotypes rather than directly sequenced variations. Causal variants or SNPs in known loci may not be represented on 1000 Genomes panels or not well imputed from reference data because of differences in linkage disequilibrium (LD) across populations. To address this limitation, we conducted WGS association analyses to identify rare, low-frequency, and genetic ancestry-specific genetic variants associated with BMI, using data from the Trans-Omics for Precision Medicine (TOPMed) Program[8], which is the most racially and ethnically diverse WGS program to date, as well as the Centers for Common Disease Genomics (CCDG) Program[9].

e-mail: xinruo@email.unc.edu; aejustice1@geisinger.edu

## Results

### Single-variant analyses

Our study population was racially, ethnically, geographically, and ancestrally diverse. We analyzed a multi-population sample of 88,873 adults from 36 studies in the Freeze 8 TOPMed and CCDG programs (Fig. 1, Supplementary Fig. 1, Supplementary Data 1–5). Among the 90 million SNPs included in the multi-population analysis, 86% ($N = 77,178,487$) were rare SNPs with a study-wide MAF of 0.5% <MAF ≤ 1%, and 6% ($N = 5,542,150$) were low-frequency (1% <MAF ≤ 5%) SNPs. In the multi-population unconditional analysis, we

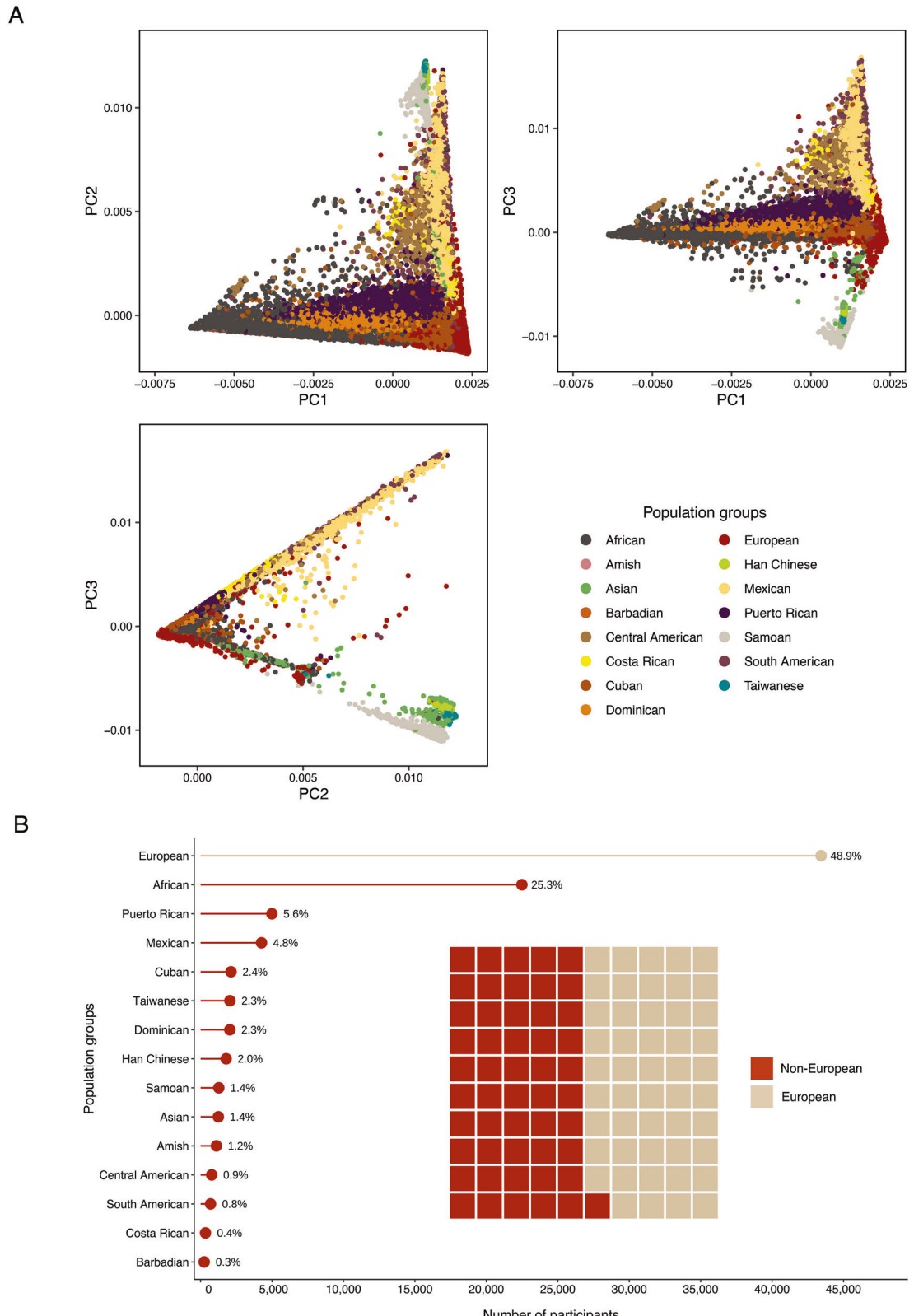

**Fig. 1 | Study population group composition. A** Pairwise scatter plots of the first three principal components (PCs) by population group. **B** The number and proportion of participants by population group. Our study population was composed of 88,873 participants from 15 population groups, 51% of which are non-European.

**Table 1 | Summary of independent loci reaching genome-wide significance (two-sided $P < 5 \times 10^{-9}$ to account for multiple testing) in single variant and internal conditional analyses**

| CHR | POS (hg38) | Nearest gene | rsID | REF | ALT | ALT Freq | Beta | SE | P-value | Known index variant[a] | Novel Locus[b] |
|-----|-----------|--------------|------|-----|-----|----------|------|-----|---------|-----------------|-------------|
| **Top variant in each locus** | | | | | | | | | | | |
| 1 | 177920345 | *SEC16B* | rs543874 | A | G | 20% | 0.064 | 0.006 | 1.38E-26 | Yes | No |
| 2 | 621558 | *TMEM18* | rs939584 | C | T | 85% | 0.058 | 0.007 | 1.99E-17 | No | No |
| 2 | 24927427 | *ADCY3* | rs10182181 | A | G | 56% | 0.035 | 0.005 | 1.76E-11 | Yes | No |
| 3 | 186108951 | *ETV5* | rs869400 | T | G | 82% | 0.038 | 0.006 | 1.21E-09 | No | No |
| 4 | 45179317 | *GNPDA2* | rs12507026 | A | T | 36% | 0.045 | 0.005 | 9.55E-19 | Yes | No |
| 5 | 75707853 | *POC5* | rs2307111 | T | C | 55% | -0.032 | 0.005 | 7.43E-10 | Yes | No |
| **6** | **50830813** | ***TFAP2B*** | **rs2206277** | **C** | **T** | **19%** | **0.054** | **0.006** | **2.05E-18** | **Novel** | **No** |
| 8 | 76068626 | *HNF4G* | rs830463 | A | G | 47% | 0.031 | 0.005 | 6.58E-10 | No | No |
| **11** | **27657463** | ***BDNF*** | **rs3838785** | **GT** | **G** | **58%** | **-0.030** | **0.005** | **3.14E-09** | **Novel** | **No** |
| 12 | 49853685 | *BCDIN3D* | rs7138803 | G | A | 30% | 0.036 | 0.005 | 1.69E-11 | Yes | No |
| 13 | 53533448 | *OLFM4* | rs9568868 | G | T | 14% | 0.047 | 0.007 | 5.73E-11 | No | No |
| 16 | 53767042 | *FTO* | rs1421085 | T | C | 29% | 0.090 | 0.006 | 6.11E-59 | Yes | No |
| 18 | 60161902 | *MC4R* | rs6567160 | T | C | 21% | 0.053 | 0.006 | 8.22E-19 | Yes | No |
| 19 | 47077985 | *ZC3H4* | rs28590228 | C | T | 50% | 0.033 | 0.005 | 4.75E-10 | No | No |
| **22** | **29906934** | ***MTMR3*** | **rs111490516** | **C** | **T** | **4%** | **0.078** | **0.013** | **4.52E-09** | **Novel** | **Yes** |
| X | 31836665 | *DMD* | rs1379871 | G | C | 41% | 0.029 | 0.004 | 1.35E-11 | Yes | No |
| **Secondary signals** | | | | | | | | | | | |
| 2 | 422144 | *ALKAL2* | rs62107261 | T | C | 3% | -0.095 | 0.014 | 3.83E-12 | Yes | No |
| 18 | 60361739 | *MC4R* | rs78769612 | G | T | 2% | -0.106 | 0.019 | 3.53E-08 | No | No |

Newly identified locus highlighted in bold.

*CHR* chromosome, *POS* position, *REF* reference allele, *ALT* alternative allele, *ALT Freq* alternative allele frequency, *SE* standard error.

[a]Known index variant 'Yes' indicates a previously published index variant from NHGRI-EBI GWAS Catalog; 'No' indicates index variant within 500 kb ± of the published lead variant, not independent of known signal in conditional analysis; 'Novel' indicates new lead variant either not published or conditionally independent.

[b]Novel locus 'Yes' was defined if there is no known index variant within 500 kb ± of the lead variant in the current analysis.

identified 16 loci that reached the prespecified genome-wide significance threshold of $P < 5 \times 10^{-9}$ (Table 1, Fig. 2, Supplementary Figs. 2 and 3), including one low-frequency (MAF = 4%) and 15 common (MAF 14%–50%) tag SNPs. In general, the low-frequency variant in our primary discovery showed a stronger effect than the common variants, with an estimated effect 2.14 times larger than the average common variants (0.078 *vs*. 0.037 on average). Of these 16 loci, 15 were in known BMI-associated regions, and one novel locus was identified on chromosome 22 harboring a low-frequency index SNP near *MTMR3* (Myotubularin-Related Protein 3; rs111490516; MAF = 4%, β = 0.078, SE = 0.013, $P = 4.52 \times 10^{-9}$; Table 1). The MAF of this *MTMR3* locus varied widely across population groups, with the highest MAF observed in the African (13%) and Barbadian (13%) population groups, while it ranged from 0% to 5% in other population groups (Supplementary Data 5).

In the two population-specific analyses, 10 association signals reached genome-wide significance (Supplementary Data 6, Supplementary Figs. 4–7). All 10 association signals were identified in the multi-population analysis, including two of these loci with the same index SNPs in at least one population, *SEC16B* in Africans and *FTO* in Europeans. For these same two loci, each population-specific analysis also revealed a distinct lead variant compared to the multi-population analysis; however, they were in high LD with ($R^2 = 0.95$ and $R^2 = 1.00$, according to TOP-LD[10]; Supplementary Data 6) and within 30 kb of the multi-population lead SNPs. Notably, the novel locus in *MTMR3* achieved significance exclusively in the African group. While the most significant SNP in the African population group (rs73396827) differed from that in the multi-population analysis (rs111490516), the two were in strong LD in the TOPMed African population ($R^2 = 1.00$). Both of these SNPs were fixed in the European group (MAF = 0%). Our sensitivity analysis illustrates the robustness of our findings as we identified the same three significant association

signals ($P < 5 \times 10^{-9}$) and lead SNPs as in the main analysis (Supplementary Data 7). Additionally, there was no substantial change in the effect estimate (directionally consistent and sensitivity |Beta| > 90% original |Beta|). In the European group analysis, one SNP in the *ALKAL2* locus on chromosome 2 (rs62107261, β = −0.102, SE = 0.016, $P = 2.08 \times 10^{-10}$, MAF = 5%) was not in LD with the corresponding lead variant in the multi-population analysis ($R^2 = 0.00$, as calculated in the analysis subset), but was a known independent secondary signal at this locus[11]. The remaining SNPs were in the proximity to and in LD with the index SNPs in the corresponding loci from the multi-population analysis.

## Replication

The replication sample sizes ranged from 3213 in MyCode to 80,730 in MVP (Supplementary Data 8). In the six replication studies of Black, African, and African American participants, rs111490516 and rs73396827 in *MTMR3* displayed high LD ($R^2 = 1.00$) (Supplementary Data 8). Their MAFs are nearly identical across studies and ranged from 11% to 13%, aligning with the 13% observed in our African and Barbadian groups (Supplementary Data 8). We replicated both variants, demonstrating directionally consistent associations with BMI across the replication studies. In MVP and REGARDS, particularly, both variants reached statistical significance at $P < 0.05$. We observed a 68% reduction in the estimated effects when meta-analyzing across replication studies (rs111490516: β = 0.025, SE = 0.007, $P = 1.92 \times 10^{-4}$, MAF = 12%; rs73396827: β = 0.026, SE = 0.007, $P = 1.50 \times 10^{-4}$, MAF = 12%), compared to the discovery analysis (Fig. 3, Supplementary Data 8, Supplementary Fig. 8). In the meta-analysis of up to 202,675 individuals from both discovery and replication studies, rs111490516 and rs73396827 both had an estimated effect of 0.36 with a SE of 0.006 and P-values of $2.40 \times 10^{-9}$ and $1.60 \times 10^{-9}$, respectively (Fig. 3, Supplementary Data 8).

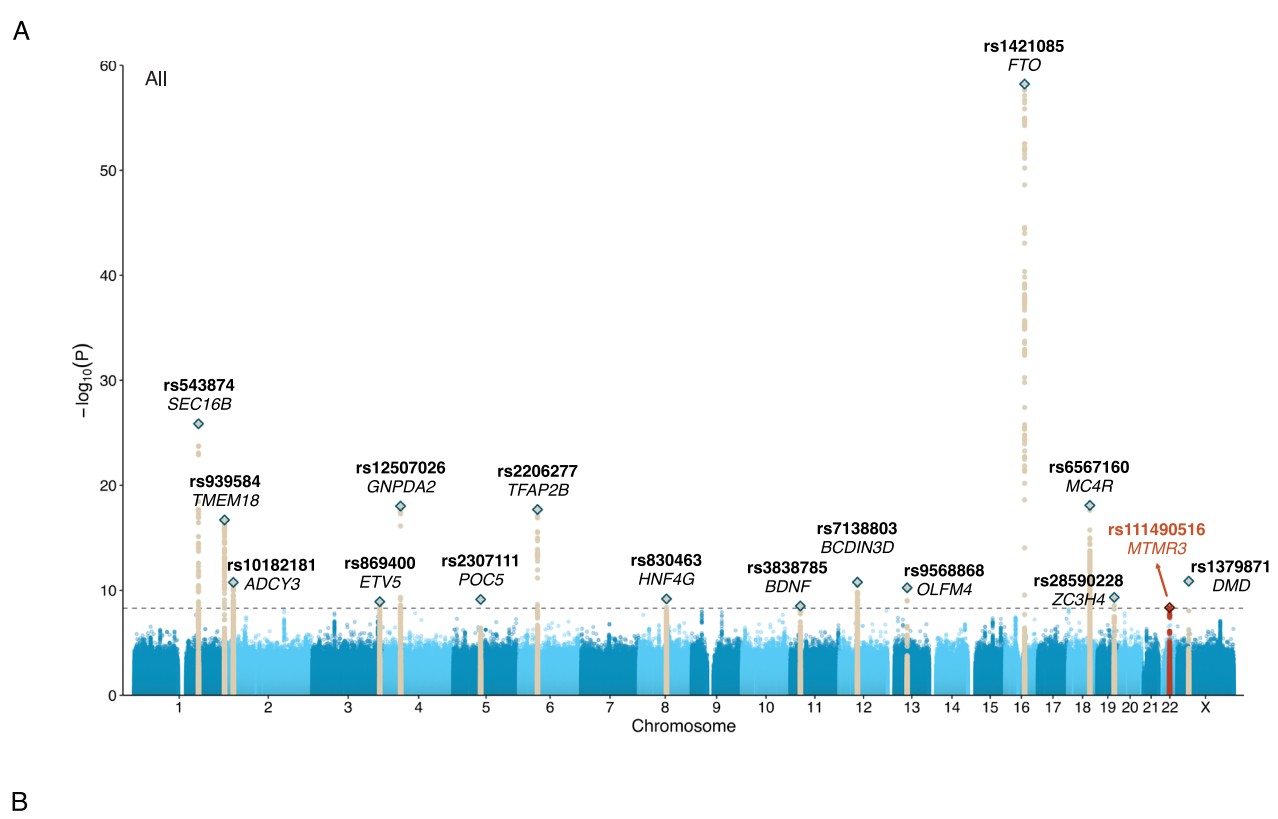

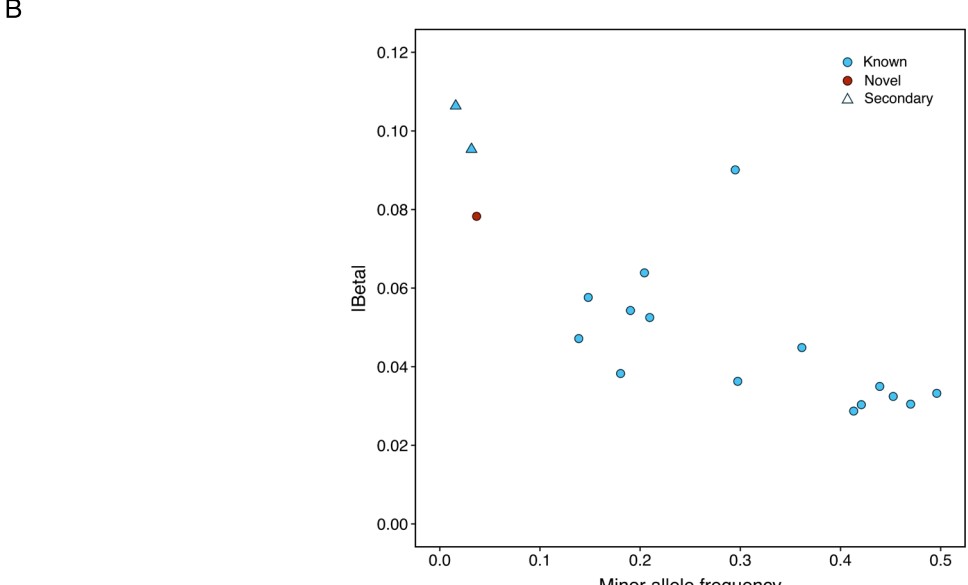

**Fig. 2 | Summary of significant association findings. A** Manhattan plot of multi-population, single-variant analysis ($N$ = 88,873 individuals). The novel locus *(MTMR3)* is highlighted in red. Previously reported BMI loci are in dark beige. The horizontal dashed line indicates the genome-wide significance threshold two-sided $P = 5 \times 10^{-9}$, to account for multiple testing. **B** Scatterplot showing the minor allele frequency compared to the absolute value of the estimated effect of the index variant at each significant locus. All effect estimates are from the primary analysis conducted across all population groups. Previously reported loci are highlighted in blue, while the novel locus is in red; circles represent the most significant variant at each locus, and triangles show newly reported secondary signals within known loci.

## Functional annotation

To gain a better understanding of the potential functional consequence of the *MTMR3* locus, we used publicly available information from Ensembl Variant Effect Predictor (VEP)[12], FORGEdb[13], and ENCODE[14,15] to annotate nearby variants. Of the 54 variants in high LD with our lead SNP, most were intronic or nearby *MTMR3* (Supplementary Data 9). Of these, four variants had a moderate CADD (combined annotation dependent depletion) score (scaled CADD > 10) with rs73394881 having the highest relative CADD score[16], three of which lay within a possible enhancer (rs73396896, $R^2$ = 0.884; rs73394881, $R^2$ = 0.889; rs74832232, $R^2$ = 0.889). There is an abundance of evidence for regulatory action in this region for variants in high LD with our lead SNP. Although our top associated SNP does not directly overlap potential candidate cis-regulatory elements (cCREs) based on ENCODE, there is limited data of overlapping cCREs from RoadMap data available on FORGEdb (Supplementary Fig. 9, Supplementary Data 9). Three genes were implicated across these databases, including *MTMR3*, *MTFP1* (Mitochondrial Fission Process Protein 1), and *ASCC2*

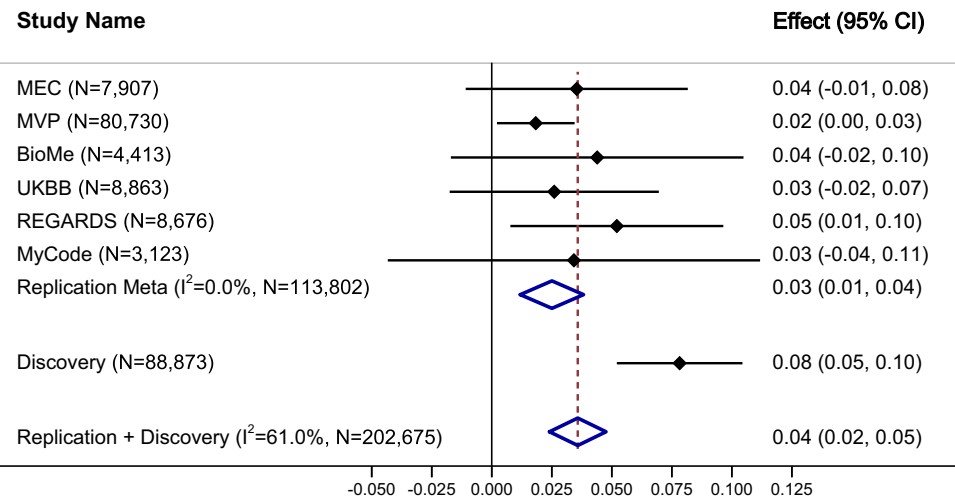

**Fig. 3 | Forest plot of rs111490516 replication.** The forest plot, centered on effect estimates with 95% confidence intervals, is oriented on the BMI-increasing allele. Effect estimates are provided as standard deviation in BMI per allele. Standard errors and *P*-values of the effect estimates are provided in Supplementary Data 8.

(Activating Signal Cointegrator 1 Complex Subunit 2). Five SNPs were implicated in potential regulation for *MTMR3* and/or *ASCC2* across multiple tissues based on Activity-By-Contact (ABC) data. Additionally, 35 SNPs were significant cis-eQTLs with *MTFP1* in Nerve Tibial cells. Three SNPs exhibited Zoonomia PhyloP scores indicative of accelerated evolution (PhyloP ≤ −2.270), including our tag SNP, rs111490516, and two nearby SNPs, rs57349783 and rs2107673; meanwhile, three exhibited scores indicative of significant cross-species conservation (PhyloP ≥ 2.270), including rs73396896, rs6006286, and rs73394881[17].

**Conditional analyses**

Conditional analysis using the most associated variant at each locus revealed two significant secondary signals after multiple testing correction (Table 1, Supplementary Data 10, Supplementary Fig. 10). These included a known BMI-associated index variant on chromosome 2 (rs62107261, β = −0.097, SE = 0.014, *P* = 2.06 × 10⁻¹², near *ALKAL2*)[11], which was also the most significant variant at this locus in the European group analysis (Supplementary Data 6). We further identified rs78769612 on chromosome 18 (β = −0.100, SE = 0.019, *P* = 2.17 × 10⁻⁷, near *MC4R*). Although both secondary SNPs were in known BMI-associated loci, rs78769612 near *MC4R* was a new index variant.

We additionally assessed independence for the top variants in known loci by conditioning on all previously reported index variants by chromosome[5,11,18–33], and highlighted previously reported index variants located within 500 kb of our top locus SNP (Supplementary Data 11). Two SNPs, rs2206277 in *TFAP2B* and rs3838785 in *BDNF*, remained significant after multiple test corrections, indicating potentially novel signals in known loci (Supplementary Data 11). The novel index variant from the internal conditional analysis, rs78769612 near *MC4R*, was not robust to this treatment, suggesting that this novel variant was not independent of known BMI variants. The LD matrix plots highlighted low LD (*R²* range 0.018–0.342) between our top SNP at the *BDNF* locus, rs3838785, and previously published lead variants within 500 kb (Supplementary Fig. 11). Although our top SNP, rs2206277, in the *TFAP2B* locus was conditionally independent of previously published BMI-risk SNPs (β ≥ 90% of the unconditioned β and *P* < 6.25 × 10⁻³), this SNP was in high LD with two nearby published SNPs (*R²* = 0.822 for rs987237 and *R²* = 0.793 for rs72892910).

**Aggregate-based testing**

We did not identify any novel gene regions through association tests at the genome-wide level (*P* < 5 × 10⁻⁷) when aggregating variants with MAF ≤ 1%. Nevertheless, we successfully replicated previous gene-

based associations with the well-known *MC4R* gene (*P* = 8.47 × 10⁻⁸), with 110 alleles across 37 sites within coding regions, enhancers, and promoters for *MC4R*. The *MC4R* locus was also identified in single-variant analyses. We also examined the 16 genes reported by Akbari et al.[34] with an exome-wide significant association with BMI (Supplementary Data 12). Although only *MC4R* reached genome-wide significance in our aggregate-based testing, three additional genes – *ROBO1*, *GPR151*, and *ANO4* – showed nominal significance (*P* < 0.05) in our SMMAT test.

**Fine-mapping**

To pinpoint the most probable causal variant(s) underlying each of the 16 loci, we subsequently performed fine-mapping using PAINTOR[35]. Assuming one causal variant per locus, the index variants were the most likely causal variants in 14 loci, with posterior probability (PP) ranging from 0.02 and 1.00, and seven of them had a PP above 0.50 (Supplementary Data 13, Supplementary Fig. 12). Two intronic index variants, rs2307111 in *POC5* and rs1379871 in *DMD*, were particularly noteworthy with PP exceeding 0.98. In contrast, variants with the highest PP in *ADCY3* and *ZC3H4* were not the reported index variants, although the highest PP for the *ADCY3* locus was below 0.50 and thus not likely the causal variant underlying this signal. In the *ZC3H4* locus, the highest PP variant (rs55731973, PP = 0.77) was intronic, located in the 5′ UTR or upstream of *ZC3H4* depending on alternative transcripts, and resided in probable enhancer regions. Additionally, this variant was a significant cis-eQTL for *SAE1*[36], another nearby downstream gene. Two of our loci identified potential secondary signals following conditional analysis, the *TMEM18/ALKAL2* and *MC4R* loci, but both secondary signals were >100 kb from our index variant.

Of the seven variants with PP above 0.50, all have evidence of moderate to high impact on a nearby gene. All had FORGEdb[13] scores of 5 or above, and six had CADD scores greater than 15[16] for at least one transcript (Supplementary Data 9). For example, the causal variant implicated in our fine-mapping for the *FTO* locus, rs1421085, is intronic or within an enhancer for *FTO*, has a FORGEdb score of 9, and scaled CADD score of 19.58. Multiple studies on this known obesity-risk variant have confirmed this as a causal variant in the pathogenesis of obesity[37–40]. Also, rs55731973, in the *ZC3H4* locus, while not our index SNP, has a FORGEdb score of 10 (the highest score) and a scaled CADD score of 18.84 due to multiple lines of evidence that it sits in an active regulatory site across multiple tissues. None of the seven exhibited a CATO (Contextual Analysis of TF Occupancy) score >0.1, indicative of strong cross-tissue evidence of transcription regulation[41], a

conservative metric for identifying high-confidence regulatory variants. Four of the seven SNPs exhibited high Zoonomia PhyloP scores indicative of significant cross-species conservation (PhyloP ≥ 2.270), including rs1379871, rs2307111, rs55731973, and rs1421085[17].

## PheWAS

To explore potential novel pleiotropy, we conducted association tests between the tag variant from our novel locus, rs111490516, and 538 PheCodes available in the MyCode and BioMe studies. No PheCode was significantly associated with rs111490516 following multiple test corrections ($P < 9.3 \times 10^{-5}$). However, PheCode 327.3 (Sleep Apnea) and 327.32 (Obstructive Sleep Apnea) ranked among the top associated PheCodes ($P < 0.001$) (Supplementary Data 14, Supplementary Fig. 13). Perhaps not coincidentally, obesity is one of the strongest risk factors for sleep apnea[42].

## Discussion

By leveraging WGS data from a large multi-population study, we identified and replicated one novel low-frequency BMI variant in *MTMR3* specific to the diversity of our sample. We also identified two common secondary signals in known BMI loci, supported gene-based associations for *MC4R*, and refined resolution in multiple loci by prioritizing candidate SNPs with high PP. Our discovery of the novel BMI-associated variant emphasizes the importance of studying diverse populations, which could further refine and expand the catalog of genes and variants that confer risk for obesity and potentially other disease traits.

The novel *MTMR3* variant, rs111490516, was most common in our African and Barbadian population groups (MAF = 13%) and of moderate frequency in our Dominican population group (MAF = 5%). We further replicated this association in study samples of similar population backgrounds. Yet, previous GWAS of BMI focusing on African ancestry individuals failed to identify a significant association in this region. It is not available for lookup in the most recently published MVP BMI GWAS[43], although included in our replication with the latest MVP data release. In one of the largest GWAS meta-analyses of imputed genotype data in African ancestry individuals with summary data available publicly, which was conducted by the African Ancestry Anthropometry Genetics Consortium (AAAGC, N up to 42,751)[23], this variant was directionally consistent and suggestively associated (β = 0.042, $P = 1.80 \times 10^{-4}$, MAF = 12%)[23]. Similarly, in our replication analysis of 113,802 individuals with imputed genotypes, rs111490516 was suggestively significant (β = 0.025, $P = 1.92 \times 10^{-4}$, MAF = 12%). In the African-specific analysis, although rs11490516 was genome-wide significant (β = 0.087, SE = 0.015, $P = 1.82 \times 10^{-9}$), rs73396827 was the top hit at this locus. These two SNPs further exhibited absolute LD, as well as consistent directions of effect, SEs, and $P$-values across replication studies and in the meta-analyses. Therefore, the lack of discovery in prior publications may not be due to insufficient power. As indicated by our fine-mapping results and the potential regulatory role of nearby variants in this novel locus, our index SNP is likely not causal but could be in LD with a causal SNP and also poorly captured in studies relying on imputation. In other words, the causal variant underlying this locus may be nearby, less frequent, and on an LD block more frequent in a population poorly represented in other imputation reference panels but well represented in our WGS and highly diverse sample (e.g., Caribbean admixed individuals). The non-European-ancestry populations particularly fall short in imputation performance due to their persistent underrepresentation in reference panels and overestimate of imputation accuracy[44]. In this case, one would require sequencing data in a large sample size with the relevant haplotype to detect a significant association that was not able to be identified with imputation in a similar number of people. Unfortunately, we are unable to test this hypothesis in our data due to data access constraints. Thus, future studies with WGS in study populations with genetic similarity

and functional follow-work are needed to further narrow in on the causal variant(s) underlying this association signal.

The SNP rs111490516 lies in an intron of the *MTMR3* (myotubularin-related protein 3) gene, with limited evidence of involvement in regulatory or functional protein activity. Other variants mapped to *MTMR3* have been associated with obesity-related traits in GWAS. In a study of 155,961 healthy and medication-free UKBB participants, rs5752989 near *MTMR3* was associated with fat-free mass (β = 0.115, $P = 8.00 \times 10^{-9}$, allele G frequency = 43%)[45]. In a meta-analysis of up to 628,000 BioBank Japan (BBJ), UKBB, and FinnGen (FG) participants, the same SNP was associated with body weight (β = -0.010, $P = 3.86 \times 10^{-8}$, allele A frequency ranged from 51% in FG to 86% in BBJ)[46]. However, this SNP (rs5752989, chr22:29969791) is not in LD of our top associated SNP in this region in our study population ($R^2 = 0.03$), and thus likely an independent signal.

While there is limited knowledge on the biological implications of the lead variant, there is an abundance of evidence for a regulatory role for SNPs in high LD with our lead SNP and multiple lines of evidence supporting a role in obesity at this locus. We utilized the Ensembl VEP, FORGEdb, and ENCODE databases to explore the predicted functional consequences of our novel locus. Our lead variant is intronic to *MTMR3*, and there is evidence linking SNPs in high LD with the regulation of *MTMR3*, *MTFP1*, and *ASCC2*. The primary cellular function of *MTMR3* relates to the regulation of autophagy[47]. Although there is no direct evidence linking *MTMR3* to obesity, previous studies have established a connection between *MTMR3* and related cardiometabolic traits. *MTMR3* was associated with LDL cholesterol ($P = 1 \times 10^{-8}$) in a GWAS meta-analysis of European, East Asian, South Asian, and African ancestry individuals[48]. A potential mechanism was proposed later suggesting *MTMR3* may mediate the association between miRNA-4513 and total cholesterol[49]. Furthermore, pyruvate dehydrogenase complex-specific knockout mice with high-fat diet-induced obesity also exhibited increased blood glucose and higher expression levels of *MTMR3*[50]. *ASCC2* has no known role related specifically to obesity. However, like previously implicated BMI-associated genes[5], it is a ubiquitin-binding protein involved in transcriptional regulation and DNA repair[51–54]. There is strong evidence for a role of MTFP1 in the regulation of adipogenesis from animal models. For example, liver-specific knockout of MTFP1 in mice provides evidence for protection against weight gain and ensuing diseases of metabolic dysregulation (e.g., diabetes and non-alcoholic fatty liver disease)[55]. The protection against weight gain appears specific to lipid-rich diets through and less pronounced in exposure to carbohydrates. Furthermore, the knockdown of *MTFP1* in adipocytes from sheep increased adipogenesis. Decreased expression of *MTFP1* was linked to increased expression of *PPARG* (Peroxisome Proliferator-Activated Receptor Gamma) and *LPL* (Lipoprotein Lipase), two known obesity-related genes important for adipogenesis, lipid metabolism, and ultimately energy homeostasis. Also, differential expression of *MTFP1* was observed during increased adipogenesis and fat deposition in the tail of developing sheep[56].

The use of WGS, coupled with the inclusion of non-European populations, improved fine-mapping resolution, as has been shown previously[32]. While there have been multiple attempts to fine-map previously identified BMI loci[5,20,33], no previous peer-reviewed and published study has successfully identified BMI-risk variants of high confidence at the *POC5* and *DMD* loci. By applying a Bayesian fine-mapping approach, we reduced associated signals to 95% credible sets of two likely causal SNPs. Functional annotation revealed that one of them, rs2307111, was a benign missense variant in *POC5* (NP_001092741.1:p.His36Arg) according to ClinVar[57,58], while the other is an intron variant in the promotor region of *DMD*. Nevertheless, these two variants were also considered high-confidence causal variants ($PP_{rs2307111} = 0.96$ in UKBB, $PP_{rs1379871} = 0.99$ in both UKBB and FG) in a recent preprint of a joint analysis of three biobanks (UKBB, FG, BBJ)[59]. Notably, unlike in Kanai et al. where the PP appeared to be driven by

the Europeans (for rs2307111: $PP_{UKBB} = 0.96$, $PP_{BBJ} = 0.12$, $PP_{FG} = 0.01$; for rs1379871: $PP_{UKBB} = 1.00$, $PP_{FG} = 1.00$), the effect alleles in our study were observed in high proportions across many non-European population groups (Supplementary Data 5). Beyond these two high confident SNPs, five additional SNPs in as many loci were identified in our fine-mapping analysis with a PP > 0.5, all with moderate to high evidence of a regulatory or functional role related to a nearby gene. Given that one of these SNPs, rs1421085 in *FTO*, has been successfully confirmed as a causal variant at this locus[37–40], additional variants highlighted in our fine-mapping analysis warrant consideration in future functional studies. While the use of population-matched LD is ideal for trans-population fine-mapping, such data remain scarce in non-European populations[60]. Although trans-population fine-mapping strategies assuming multiple causal variants are considered superior[61], PAINTOR may yield unreliable results under certain conditions, even when assuming a single variant[62]. Even though we are encouraged that Kanai et al.[59], using FINEMAP[63] and SuSiE[64], identified the same two variants as high-confidence causal variants, cross-validation of results using other fine-mapping approaches remains necessary.

In addition to our novel findings, 17 of the 18 identified variants reside in previously reported BMI-associated loci, highlighting the generalizability of the genes underlying BMI across populations, including *SEC16B*, *TMEM18*, *ETV5*, *GNPDA2*, *BDFN*, and *MC4R*[5,21,33,65]. Three of the loci harbor genes implicated in severe and early-onset obesity – *ADCY3*, *BDNF*, and *MC4R*[4]. We also consistently identified multiple association signals of high effect in *MC4R*, which is a well-established monogenic obesity gene, through our discovery analysis, internal conditional analysis, and rare variant aggregate analysis. Despite not identifying novel SNPs in *MC4R* that are independent of known BMI-associated SNPs, we replicated a secondary signal in this gene, rs2229616, a rare missense variant previously reported in individuals of European ancestry by Speliotes et al.[66]. In addition to *MC4R*, three other genes – *ROBO1*, *GPR151*, and *ANO4* – of the exome-wide significant genes identified in Akbari et al.[34] showed nominal significance in our rare variant aggregate analysis. This lends further support to the generalizability of these genes across populations, given that 85% were of European ancestry and 15% of admixed American ancestry in Akbari et al., compared to our more diverse cohort with 49% European and 51% other populations. We did not replicate 12 of the gene-based findings from Akbari et al. The previous study had a nearly seven-fold larger sample size compared to the current study, included related individuals increasing the opportunity for multiple copies of rare variants, and implemented alternative methods for gene-based analysis and variant binning. Therefore, it is likely a combination of power due to differences in sample size, underlying variant selection, difference in methods for gene-based analysis, and potential winner's curse that contributed to the lack of validation of their findings in the current study.

While our study included a large sample size of diverse populations and leveraged high-quality WGS data from well-characterized and harmonized cohorts, our results should also be interpreted with the following limitations. First, although our study is large compared to other harmonized and sequenced data samples, the total study size is relatively modest compared to existing GWAS meta-analyses of common variants using imputed genotype data. Moreover, rare variants, such as those analyzed in our study, may require even larger sample sizes for novel discoveries. Even though our study is among the most racially, ethnically, and ancestrally diverse yet conducted, the European population group still represented 49% of our participants. On the other hand, diversity can contribute to added heterogeneity of effect sizes for common and rare variants, potentially limiting discovery in the multi-population analysis. We sought to overcome this limitation by allowing for heterogeneous residual variances across

population groups and examining population-stratified results when sample sizes were adequate. Notably, all our genome-wide significant loci from population-stratified analyses were also captured in the multi-population analysis, likely owing to our considerations of heterogenous effects, self-identity (population groups), and genetic ancestry (genotype principal components [PCs]). As has been shown by others[32,67,68], this underscores the importance of conducting multi-population analysis using appropriate methods that account for heterogeneity and minimize the risk of inflation or missed detection of loci that may vary in MAF or phenotypic effects across populations.

In summary, our study demonstrates the power of leveraging WGS data from diverse populations for new discoveries associated with BMI. As we enter the era of incorporating GWAS-based risk models in clinical practice, it is critical that we continue to diversify the data collected and analyzed in genomic research. Failure to do so risks further exacerbating health disparities for public health crises such as obesity. Ultimately, our study brings us one step closer to understanding the complex genetic underpinnings of obesity, translating these leads into mechanistic insights and developing targeted preventions and interventions to address this global public health challenge.

## Methods

### Study population and phenotype

Our study population was racially, ethnically, geographically, and ancestrally diverse. We analyzed a multi-population sample of 88,873 adults from 36 studies in the freeze 8 TOPMed and CCDG programs (Fig. 1, Supplementary Data 1). TOPMed Program individual-level data is available through Google and AWS cloud services following NIH dbGap approval. Details on gaining access are found on the TOPMed website (see Data Availability Statement). They belonged to 15 population groups, reflecting the way participants self-identified in each study. For individuals who had unreported or non-specific population memberships (e.g., "Multiple" or "Other"), we applied the Harmonized Ancestry and Race/Ethnicity (HARE) method[69] to infer their group memberships using genetic data, excluding these individuals from the training step. This imputation was applied to 8015 participants (9% of the overall population), assigning each to one of the existing population groups based on the group with the highest probability of membership. All other participants remained in the population group assigned based on their self-reported race/ethnicity/population group. In this way, our study population groups were defined based on a combination of self-reported identity and the first nine genetic PCs (Fig. 1, Supplementary Fig. 1, and Supplementary Data 1). The decision to use nine PCs was informed by the elbow method and scree plots.

The 15 population groups were labeled by their self-identified or primary inferred population group (e.g., predominantly African ancestry/admixed African/Black participants were labeled as "African"). Sample sizes for these groups ranged from 341 to over 43,000 as follows: African (N = 22,488), Amish (N = 1106), Asian (N = 1241), Barbadian (N = 248), Central American (N = 776), Costa Rican (N = 341), Cuban (N = 2128), Dominican (N = 2046), European (N = 43,434), Han Chinese (N = 1787), Mexican (N = 4265), Puerto Rican (N = 4991), Samoan (N = 1274), South American (N = 695), and Taiwanese (N = 2053). We refer to analyses involving all 15 population groups as multi-population analyses and group-specific analyses by their primary population group.

Among the 88,873 participants, 53,109 (60%) were female, and 45,439 (51%) were non-European. The mean (SD) age of the participants was 53.5 (15.1) years. Additional descriptive tables of the participants are presented in Supplementary Data 2–4. BMI was calculated by dividing weight in kilograms by the square of height in meters. Participants were excluded from analyses if less than 18 years of age,

had known pregnancy at the time of BMI measurement, had implausible BMI values (above 100 kg/m$^2$ without corroborating evidence), or did not provide appropriate consent. The mean (SD) of BMI varied by study, ranging from 23.4 (3.1) in GenSALT to 33.9 (7.8) in DHS (Supplementary Data 2), and by population group, ranging from 23.4 (3.1) in Han Chinese to 33.7 (6.8) in Samoans (Supplementary Data 3).

## TOPMed WGS

In TOPMed, ~30× WGS was conducted using Illumina HiSeq X Ten instruments at six sequencing centers[8]. At the Center for Statistical Genetics at the University of Michigan, TOPMed sequence data were mapped to the GRCh38 human genome reference sequence in a manner consistent with the joint CCDG/TOPMed functionally equivalent read mapping pipeline[70]. Joint genotype calling on all samples in Freeze 8 used the GotCloud pipeline[71]. Variants were filtered using a Support Vector Machine (SVM) implemented in the libsvm software package. Sample-level quality assurance steps included concordance between annotated and genetic sex, between prior SNP array genotyping and WGS-derived genotypes, and between observed and expected relatedness and pedigree information from cleaned sequence data. These details regarding the laboratory methods, data processing, and quality control are also described on the TOPMed website (https://topmed.nhlbi.nih.gov/topmed-whole-genome-sequencing-methods-freeze-8).

## Common variant association analysis

We performed a multi-population WGS association analysis of BMI using GENESIS[72] on the Analysis Commons (http://analysiscommons.com)[73] computation platform. GENESIS was chosen due to its analytical flexibility in relationship to allowing for heterogeneity of effect by population group[72], an option well-suited to the demographic and genetic background of our study population. Analyses were performed using linear mixed models (LMM). To improve power and control for false positives with a non-normal phenotype distribution, we implemented a fully adjusted two-stage procedure for rank-normalization when fitting the null model[74]. The first model was fit by adjusting BMI for age, age squared, sex, study, population group, and genetic ancestry-representative PCs generated using PC-AiR[75], sequencing center, sequencing phase, and project. A 4th-degree sparse empirical kinship matrix (KM) computed with PC-Relate[67] was included to account for genetic relatedness among participants. We also allowed for heterogeneous residual variances across sex by population group (e.g., female European), as it has previously been shown to improve control of genomic inflation[76]. Residuals from the first model were rank-normal transformed within population group and sex strata. The resulting transformed residuals were used to fit the second-stage null model, allowing for heterogeneous variances by the population group and sex strata and accounting for relatedness using the kinship matrix. Variants with a MAF of at least 0.5% were then tested individually. Due to the large number of variants tested ($N = 90{,}142{,}062$) in the multi-population analysis, we adopted a significance threshold of $5 \times 10^{-9}$ as has been used previously[77]. This approach maximizes participant inclusion, thereby maximizing the sample size in our discovery cohort to increase statistical power and avoid the misinterpretation of group-specific effects in underpowered strata. By including ancestrally diverse populations, we leverage differences in LD across populations, which has been shown to help identify novel loci, narrow down causal variants, and improve the variance explained in models[32,68,76]. For quantitative traits like BMI, multi-population analyses that account for heteroscedasticity of genetic effects across population groups have proven effective in increasing study power and reducing genomic inflation[32,76]. Lastly, and most importantly, a pooled approach aids in decreasing health disparities by identifying loci that generalize across populations.

Additionally, group-specific analyses were conducted in the two largest population groups, European and African, to determine whether a particular group with a large sample size is driving the observed association signals. To address any concerns over potential residual confounding due to population substructure in our African group-specific GWAS, especially for our novel association that differed in allele frequency across population groups, we conducted a sensitivity analysis using group-specific PCs. The sensitivity analysis used 10 African group-specific PCs calculated in the same individuals as used in the pooled analysis and the same covariates as before.

## Replication analysis

For the novel single-variant association identified in the *MTMR3* locus from our discovery analyses that is largely driven by the African population group, we requested replication specifically in Black, African, and African American participants from six independent cohorts ($N_{total}$ = up to 113,802): BioMe BioBank Program[78], Million Veteran Program (MVP)[43,79], Multiethnic Cohort (MEC)[80], MyCode Community Health Initiative Study (MyCode)[81], REasons for Geographic And Racial Differences in Stroke (REGARDS) study[82], and UK Biobank (UKBB)[83,84]. Phenotypes were developed and analyses were carried out under the same protocol as outlined above. We subsequently conducted inverse variance weighted fixed effects meta-analysis in STATAv15.1[85], using study-specific summary results. Additional details on the parent study design for each replication study are included in the Supplementary Note.

## Functional annotation

To gain a better understanding of the potential functional consequence of the *MTMR3* locus, we used Ensembl VEP[12] and FORGEdb[13] to annotate all variants in high LD with our top SNP ($R^2 > 0.8$ in the African population group using TOP-LD[10]). Additionally, we looked for variants that overlap with potential cCREs within 100 kb of the lead index variants using visual inspection of regional association plots, including LD information. The resources included signatures of promoters, enhancers, and chromatin accessibility (i.e., markers of histone modification, DNase hypersensitivity, CTCF binding, etc.) from BMI-relevant tissues, including blood, brain, and liver from ENCODE[14,15].

## Conditional analysis

To identify loci harboring multiple independent signals, we performed stepwise conditional analyses on the most significant signal within 500 kb of our index variant. The significance threshold for secondary signals was determined by Bonferroni correction for the number of variants across all regions tested, $P = 5.96 \times 10^{-7}$ ($P < 0.05/83{,}928$ SNPs with MAF > 0.5% within 500 kb of the 16 index SNPs). Variants passing the significance threshold after the first round were further conditioned on the top variant in the locus after the first round of conditioning to identify potential third signals within each locus using the same threshold.

To determine whether association signals in known loci were independent of known signals, we performed conditional analyses using previously published index variants[5,11,18–24,26–33,86]. Specifically, for each lead SNP that is not a previously reported BMI-associated index variant, we conditioned on all known SNPs on each chromosome. Given that these are potential new signals in regions known to influence BMI, index variants were considered independent if the estimated effect (β) value remained ≥90% of the unconditioned β value and $P < 6.25 \times 10^{-3}$ (0.05/8 loci tested). For visual inspection and calculation of LD in reference panels used for imputation in previous GWAS studies, LDlink was used to calculate pairwise LD between potentially independent signals in known loci and produce LD heatmaps using the 1000 Genomes Global reference panel[87].

## Rare variant aggregate association analysis

Rare variants with a MAF ≤ 1% were tested in aggregate by gene unit across studies in the multi-population analysis. Variants were grouped

into gene units in reference to GENCODE v28, including both coding variants and variants falling within gene-associated noncoding elements. Coding variants included high-confidence loss of function variants annotated by LOFTEE[88] (Ensembl VEP LoF = HC), missense variants (MetaSVM score >0), and in-frame insertions/deletions or synonymous variants (FATHNMM-XF coding score >0.5). In addition to coding variants, we included variants falling within the promoter of each transcript tested. Promoter regions were defined as falling in the 5 kb region 5′ of the transcript and also overlaying a FANTOM5 Cage Peak[89]. In order to identify regulatory elements likely to be acting through the tested gene, we leveraged the GeneHancer database[90]. GeneHancer identifies enhancer regions and associates them with the specific genes they are likely to regulate, allowing us to aggregate regulatory regions by the likely target gene. GeneHancer regions were limited to the top 50% scored regions, and variants falling in these regulatory elements were further filtered to those most likely to have a functional impact (FATHMM-MKL noncoding score >0.75). Variants aggregated to gene units were tested using variant-set mixed model association tests (SMMAT)[91]. Variants were weighted inversely to their MAF using a beta distribution density function with parameters 1 and 25. Genes were considered significantly associated after Bonferroni correction for the number of genes analyzed ($P < 5 \times 10^{-7}$). These annotations were selected by the TOPMed Data Coordinating Center (DCC) as part of the centralized and harmonized annotations and were chosen to focus on annotation that was previously shown to have high agreement with other annotation resources and high prediction accuracy[92].

## Fine-mapping

In order to identify candidate functional variants underlying association regions, we performed fine-mapping analyses in our multi-population GWAS single-variant association summary statistics, using the program PAINTOR[35], which integrates the association strength and genomic functional annotation. We used the annotation file from aggregate-based testing described above under "Rare Variant Aggregate Association Analysis" to identify deleterious coding variants, variants within GeneHancer regions, and variants within gene promoter regions. We restricted this analysis to variants located within ±100 kb of the locus index variants. We calculated LD using our analysis subset of the TOPMed data. As PAINTOR may be sensitive to the misspecification of the number of causal variants[62,93,94], limiting our ability to interpret findings in the absence of evidence for more than one signal, we assumed one causal variant per locus, unless evidence of independent secondary signals within the 100 kb window was identified following conditional analysis, in which case we allowed for additional causal variants per locus. Restriction to a 100 kb ± window was applied due to the computational burden of WGS data and because, for most loci, LD decays at >25 kb[95,96]. While we extend our fine-mapping locus to 100 kb to allow for potential LD beyond 25 kb, one limitation of this approach is that we may still miss potentially causal variants that are >100 kb from our index SNP, including secondary signals. To gain a better understanding of the potential functional consequence of likely causal variants within each locus, we used Ensembl VEP[12] and FORGEdb[13] to annotate all variants with $PP > 0.5$.

## PheWAS

To identify potential novel phenotypic associations with newly discovered variants, we performed a phenome-wide association (PheWAS) in the MyCode, a hospital-based population study in central and northeastern Pennsylvania[81], and in the Charles Bronfman Institute for Personalized Medicine's BioMe BioBank Program located in New York City[97]; both studies had genetic data linked to electronic health records (EHR). ICD-10-CM and ICD-9-CM codes were mapped to unique PheCodes using the Phecode Map v1.2[98] from the EHR. Cases were defined if individuals had two or more PheCodes on separate dates, while

controls had zero instances of the relevant PheCode. We performed association analyses on PheCodes with $N \geq 20$ cases and 20 controls using logistic regressions, adjusting for current age, sex (for non-sex-specific PheCodes), and the first 15 PCs in BioMe and 10 PCs in MyCode calculated from genome-wide data, and assuming an additive genetic model using the PheWAS package[99] in R.Given the potential population-specific association of our novel locus, PheWAS were restricted to African Americans in each study: in MyCode, African Americans were identified through electronic health records (a combination of self-report and clinician-reported race/ethnicity), while in BioME the identification was based exclusively on self-report. We restricted our analyses to unrelated individuals up to the 2nd degree. Association analyses were conducted within each study, followed by inverse variance weighted fixed effects meta-analysis in METAL[100]. PheCodes were deemed statistically significant after Bonferroni correction for the number of PheCodes analyzed ($P < 0.05/538 = 9.3 \times 10^{-5}$).

## Reporting summary

Further information on research design is available in the Nature Portfolio Reporting Summary linked to this article.

## Data availability

The GWAS summary data generated in this study, including pooled, African, European, and sensitivity, have been deposited in the NHGRI-EBI Catalog of human genome-wide association studies (GWAS Catalog) database under accession codes GCST90502911 to GCST90502914 (https://www.ebi.ac.uk/gwas/downloads/summary-statistics). The raw TOPMed Program individual-level data are protected due to data privacy laws, but de-identified versions are available through Google and AWS cloud services following NIH dbGap approval. Details on gaining accessing are found on the TOPMed website (see https://topmed.nhlbi.nih.gov/topmed-data-access-scientific-community and https://topmed.nhlbi.nih.gov/topmed-whole-genome-sequencing-methods-freeze-8#access-to-sequence-data). In addition to raw data and full GWAS summary statistics provided through the referenced repositories, the summary statistics on the study population used in this study, along with summary results for top findings are provided in the Supplementary Data files.

## Code availability

All protocols used for variant calling and quality control for data used in this study are described at: https://topmed.nhlbi.nih.gov/topmed-whole-genome-sequencing-methods-freeze-8. Links to relevant code, including GitHub repositories, are also provided on the same site. All GWAS analyses were performed using GENESIS, a Bioconductor package, on the TOPMed Analysis Commons. Details of the GENESIS app used on the Analysis Commons, along with underlying package information and code, are provided at: https://github.com/AnalysisCommons/genesis_wdl. All scripts used for running analyses on the TOPMed Analysis Commons using the DNAnexus Platform are available at: https://github.com/Justice-Genetics-Lab/TOPMed-WGS-BMI-GWAS/tree/main and https://doi.org/10.5281/zenodo.14708351.

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

## Acknowledgements

Individual researchers received funding for their work, including National Institute of Health (NIH) grants: R01 DK122503 (C.T.L., K.E.N., A.E.J., G.C., N.S.J.), T32 HL007055 (H.M.H.), T32 HL129982 (H.M.H.), R01HL142825 (H.M.H.), I01-BX003362 (K.C.), U01 HL120393 (M.P.C., B.D.He., D.J.), R01 HL68959 (M.E.G., A.E.A., M.J.T.), U01 HL072507 (J.E.H., J.H.), K08 HL136928 (B.D.Ho.), R01HL-120393 (D.J.), R01 HL119443 (S.L.R.K.), R01 HL055673-18S1 (S.L.R.K.), R01 HL92301 (N.D.P.), R01 HL67348 (N.D.P.), R01 NS058700 (N.D.P.), RO AR48797 (N.D.P.), R01 DK071891 (N.D.P.), R01 AG058921 (N.D.P.), F32 HL085989 (N.D.P.), U01 HL089897 (E.A.R.), U01 HL089856 (E.A.R.), R01 HL093093 (D.E.W.), R01 HL133040 (D.E.W.), I01 BX003340 (P.W.F.W.), I01 BX004821 (P.W.F.W.), U01 HL072524 (D.K.A.), R01 HL104135-04S1 (D.K.A.), U01 HL054472 (D.K.A.), U01 HL054473 (D.K.A.), U01 HL054495 (D.K.A.), U01 HL054509 (D.K.A.), R01 HL055673 with supplement -18S1 (D.K.A.), R01 HL104608 (K.C.B.), R01 AI132476 (K.C.B.), R01 AI114555 (K.C.B.), R01 HL104608-S1 (K.C.B.), P20 GM109036 (J.H.), KL2 TR002490 (L.M.R.), T32 HL129982 (L.M.R.), P01 HL132825 (S.T.W.), R35 CA197449 (X.Lin), P01 CA134294 (X.Lin), U19 CA203654 (X.Lin), R01 HL113338 (X.Lin), U01 HG009088 (X.Lin), R01 HL142302 (R.J.F.L.), R01 DK124097 (R.J.F.L.), R01 DK110113 (R.J.F.L.), R01 DK107786 (R.J.F.L.), R01 DK075787 (R.J.F.L.), NNF23SA0084103 (R.J.F.L.), NNF18CC0034900 (R.J.F.L.), NNF20OC0059313 (R.J.F.L.), X01 HL134588 (R.J.F.L.), R01 HG010297 (K.E.N.), U01 HG007416 (K.E.N.), R01 DK135938 (N.S.J.), R01 HL105756 (B.M.P.), NHLBI TOPMed Fellowship (X.Li), P30 CA008748 (M.D.); contracts HHSN268201800001I (M.P.C., B.D.He., D.J.), HHSN268201500014C (S.L.R.K.); American Diabetes Association (ADA) Grant 1-19-PDF-045 (H.M.H.); the General Clinical Research Center of the Wake Forest University School of Medicine, M01 RR07122 (N.D.P.); and a pilot grant from the Claude Pepper Older Americans Independence Center of Wake Forest University Health Sciences, P60 AG10484 (N.D.P.).

## Author contributions

Conducted analyses or contributed to figures and tables: X.Z., J.A.B., M.G., N.C., Zh.W., K.F., G.C., N.S.J., Q.H., Ze.W., M.M., S.G., and A.E.J. Supervised analyses: Y.V.S., L.A.C., L.A.L., C.T.L., R.J.F.L., K.E.N., and A.E.J. Contributed to the design of the current study: X.Z., J.A.B., M.G., H.M.H., N.C., H.X., L.A.C., L.A.L., C.T.L., R.J.F.L., K.E.N., and A.E.J. Contributed to the conception or design of the TOPMed program and its operations (including organization and policies of TOPMed – e.g., exec committee, working group conveners, NHLBI staff, etc.): N.C., Zh.W., D.L.D., B.D.He., J.E.H., S.L.R.K., T.N.K., J.S.P., E.A.R., R.S.V., J.W., D.K.A., K.C.B., J.H., S.R.H., B.M.P., L.M.R., S.S.R., J.I.R., N.L.S., K.D.T., L.A.C., C.T.L., R.J.F.L., K.E.N., and A.E.J. Provided phenotypic data and/ or biosamples: M.G., N.C., M.A.A., L.F.B., M.P.B., S.C., D.L.D., R.D., X.G., C.Ha., B.H., J.E.H., Y.H., R.D.J., S.L.R.K., E.M.L., L.L.M., T.N., N.D.P., M.H.P., E.A.R., S.M.R., D.M.D., R.S.V., D.E.W., J.W., L.R.W., L.R.Y., Z.T.Y., D.K.A., J.B., E.G.B., A.P.C., D.I.C., J.E.C., M.F., J.H., S.R.H., C.K., R.M., B.M.P., L.M.R., A.P.R., S.S.R., J.I.R., M.B.S., N.L.S., N.H., L.A.C., L.A.L., R.J.F.L., K.E.N., and A.E.J. Acquired WGS and/or other omics data: L.F.B., D.L.D., R.D., J.E.H., S.L.R.K., J.A.S., R.S.V., J.W., W.Z., D.K.A., J.B., E.G.B., J.E.C., M.F., J.H., S.R.H., C.K., B.M.P., A.P.R., S.S.R., J.I.R., N.L.S., K.D.T., L.A.C., R.J.F.L., K.E.N., and A.E.J. Created software, processed, and/or analyzed WGS or other study data for data summaries in this paper: X.Z., J.A.B., M.G., H.M.H., N.C., J.C.B., J.G.B., E.J.B., M.P.C., B.D.He., C.L., C.P.M., K.L.W., and A.E.J. Drafted the manuscript and revised according to co-author suggestions: X.Z., J.A.B., M.G., H.M.H., N.C., K.E.N., and A.E.J. Critically reviewed the manuscript, suggested revisions as needed, and approved the final version: X.Z., J.A.B., M.G., H.M.H., N.C., H.X., Zh.W., K.F., G.C., N.S.J., M.M., S.G., X.Li., Z.L., M.A.A., D.M.B., L.F.B., J.C.B., M.P.B., D.W.B., J.G.B., E.J.B., C.S.C., K.C., S.C., Y.C., L.C., M.P.C., D.L.D., M.D., R.D., C.E., A.E.F., B.I.F., M.E.G., X.G., C.Ha., B.D.He., B.H., J.E.H., Y.H., B.D.Ho., D.H., Q.H., C.Hw., R.D.J., D.J., R.R.K., S.L.R.K., T.N.K., E.M.L., M.L., C.L., L.L.M., M.N.M., C.P.M., A.C.M., T.N., J.O., C.J.O., N.D.P., J.S.P., J.A.P., U.P., M.H.P., D.C.R., E.A.R., S.M.R., D.M.D., J.R., C.M.S., J.A.S., H.K.T., R.S.V., Ze.W., D.E.W., D.I.C., J.W., K.L.W., L.R.W., P.W.F.W., L.R.Y., Z.T.Y., W.Z., S.Z., D.K.A., A.E.A., K.C.B., J.B., E.B., E.G.B., A.P.C., Y.I.C., J.E.C., M.F., V.R.G., J.H., S.R.H., L.H., M.R.I., C.K., R.M., B.D.M., M.N., B.M.P., L.M.R., A.P.R., S.S.R., J.I.R., M.B.S., N.L.S., K.D.T., M.J.T., S.T.W., Y.Z., N.H., Y.V.S., X.Lin, L.A.C., L.A.L., C.T.L., R.J.F.L., K.E.N., and A.E.J.

## Competing interests

B.D.Ho. receives grant support from Bayer and has received an honorarium from AstraZeneca for an educational lecture. B.M.Ps. serve on the TOPMed Steering Committee. C.J.O. is employed by Novartis Institute of Biomedical Research, Cambridge, MA. D.L.D. received grants from Bayer and honoraria from Novartis. K.C.B. is an employee of Tempus. L.M.R. and S.S.R. are consultants for the TOPMed Administrative Coordinating Center (through Westat). U.P. was a consultant with AbbVie, and her husband is holding individual stocks for the following companies: BioNTech SE – ADR, Amazon, CureVac BV, NanoString Technologies, Google/Alphabet Inc Class C, NVIDIA Corp, Microsoft Corp. XLin is a consultant of AbbVie Pharmaceuticals and Verily Life Sciences. The remaining authors declare no competing interests.

## Additional information

Xinruo Zhang [1,96] ✉, Jennifer A. Brody [2,96], Mariaelisa Graff [1,96], Heather M. Highland [1,96], Nathalie Chami [3,4,96], Hanfei Xu[5], Zhe Wang [3], Kendra R. Ferrier [6], Geetha Chittoor[7], Navya Shilpa Josyula [7], Mariah Meyer [6], Shreyash Gupta[7], Xihao Li [8,9,10], Zilin Li[11,12], Matthew A. Allison[13], Diane M. Becker[14,98], Lawrence F. Bielak [15], Joshua C. Bis [2], Meher Preethi Boorgula[16], Donald W. Bowden[17], Jai G. Broome [18,19], Erin J. Buth[18], Christopher S. Carlson[20], Kyong-Mi Chang [21,22], Sameer Chavan [16], Yen-Feng Chiu [23,98], Lee-Ming Chuang [24], Matthew P. Conomos [18], Dawn L. DeMeo[25], Mengmeng Du[26], Ravindranath Duggirala[27,28], Celeste Eng[29], Alison E. Fohner [30], Barry I. Freedman [31], Melanie E. Garrett [32], Xiuqing Guo[33], Chris Haiman[34], Benjamin D. Heavner [18], Bertha Hidalgo [35], James E. Hixson[36], Yuk-Lam Ho [37], Brian D. Hobbs[25,38], Donglei Hu [29], Qin Hui[39,40], Chii-Min Hwu [41], Rebecca D. Jackson[42,98], Deepti Jain[18], Rita R. Kalyani[43], Sharon L. R. Kardia[15], Tanika N. Kelly[44], Ethan M. Lange[6], Michael LeNoir[45], Changwei Li [44], Loic Le Marchand[46], Merry-Lynn N. McDonald [47], Caitlin P. McHugh[18], Alanna C. Morrison [48], Take Naseri[49,50], NHLBI Trans-Omics for Precision Medicine (TOPMed) Consortium*, Jeffrey O'Connell[51], Christopher J. O'Donnell [37,52], Nicholette D. Palmer[17], James S. Pankow [53], James A. Perry [54], Ulrike Peters[20], Michael H. Preuss [3], D. C. Rao[55], Elizabeth A. Regan[56], Sefuiva M. Reupena[57], Dan M. Roden [58], Jose Rodriguez-Santana[59], Colleen M. Sitlani[2], Jennifer A. Smith [15,60], Hemant K. Tiwari[61], Ramachandran S. Vasan [62], Zeyuan Wang[39], Daniel E. Weeks [63,64], Jennifer Wessel [65,66,67], Kerri L. Wiggins [2], Lynne R. Wilkens[46], Peter W. F. Wilson[40,68], Lisa R. Yanek[14], Zachary T. Yoneda[69], Wei Zhao [15,60], Sebastian Zöllner [70], Donna K. Arnett[71], Allison E. Ashley-Koch [32], Kathleen C. Barnes [16], John Blangero [72], Eric Boerwinkle[48], Esteban G. Burchard [73], April P. Carson [74], Daniel I. Chasman [75,76], Yii-Der Ida Chen [77], Joanne E. Curran [72], Myriam Fornage [48,78], Victor R. Gordeuk [79], Jiang He [44], Susan R. Heckbert [2,80], Lifang Hou[81], Marguerite R. Irvin[82], Charles Kooperberg[20], Ryan L. Minster [63], Braxton D. Mitchell [83], Mehdi Nouraie[84], Bruce M. Psaty [2,80,85], Laura M. Raffield[86], Alexander P. Reiner [80], Stephen S. Rich [87], Jerome I. Rotter [33], M. Benjamin Shoemaker[69], Nicholas L. Smith [88,89,90], Kent D. Taylor[33], Marilyn J. Telen [91], Scott T. Weiss [92], Yingze Zhang [84], Nancy Heard-Costa [93], Yan V. Sun [39,40], Xihong Lin [8,94], L. Adrienne Cupples [5,97,98], Leslie A. Lange[6,97], Ching-Ti Liu [5,97], Ruth J. F. Loos [3,4,95,97], Kari E. North [1,97] & Anne E. Justice [7,97] ✉

[1]Department of Epidemiology, Gillings School of Global Public Health, University of North Carolina at Chapel Hill, Chapel Hill, NC, USA. [2]Cardiovascular Health Research Unit, Department of Medicine, University of Washington, Seattle, WA, USA. [3]The Charles Bronfman Institute for Personalized Medicine, Icahn School of Medicine at Mount Sinai, New York, NY, USA. [4]The Mindich Child Health and Development Institute, Icahn School of Medicine at Mount Sinai, New York, NY, USA. [5]Department of Biostatistics, School of Public Health, Boston University, Boston, MA, USA. [6]Division of Biomedical Informatics and Personalized Medicine, University of Colorado School of Medicine, Anschutz Medical Campus, Aurora, CO, USA. [7]Population Health Sciences, Geisinger, Danville, PA, USA. [8]Department of Biostatistics, Harvard T.H. Chan School of Public Health, Boston, MA, USA. [9]Department of Biostatistics, Gillings School of Global Public Health, University of North Carolina at Chapel Hill, Chapel Hill, NC, USA. [10]Department of Genetics, School of Medicine, University of North Carolina at Chapel Hill, Chapel Hill, NC, USA. [11]Biostatistics and Health Data Science, Indiana University School of Medicine, Indianapolis, IN, USA. [12]School of Mathematics and Statistics and KLAS, Northeast Normal University, Changchun, Jilin, China. [13]Department of Family Medicine, Division of Preventive Medicine, The University of California San Diego, La Jolla, CA, USA. [14]Department of Medicine, General Internal Medicine, Johns Hopkins University School of Medicine, Baltimore, MD, USA. [15]Department of Epidemiology, School of Public Health, University of Michigan, Ann Arbor, MI, USA. [16]Department of Medicine, University of Colorado School of Medicine, Aurora, CO, USA. [17]Department of Biochemistry, Wake Forest School of Medicine, Winston-Salem, NC, USA. [18]Department of Biostatistics, School of Public Health, University of Washington, Seattle, WA, USA. [19]Department of Medicine, Division of Medical Genetics, University of Washington, Seattle, WA, USA. [20]Division of Public Health Sciences, Fred Hutchinson Cancer Center, Seattle, WA, USA. [21]The Corporal Michael J. Crescenz VA Medical Center, Philadelphia, PA, USA. [22]University of Pennsylvania Perelman School of Medicine, Philadelphia, PA, USA. [23]Institute of Population Health Sciences, National Health Research Institutes, Taipei, Taiwan. [24]Department of Internal Medicine, Division of Metabolism/Endocrinology, National Taiwan

University Hospital, Taipei, Taiwan. [25]Department of Medicine, Channing Division of Network Medicine, Brigham and Women's Hospital, Harvard Medical School, Boston, MA, USA. [26]Department of Epidemiology and Biostatistics, Memorial Sloan Kettering Cancer Center, New York, NY, USA. [27]Life Sciences, College of Arts and Sciences, Texas A&M University-San Antonio, San Antonio, TX, USA. [28]Department of Health and Behavioral Sciences, College of Arts and Sciences, Texas A&M University-San Antonio, San Antonio, TX, USA. [29]Department of Medicine, Lung Biology Center, University of California, San Francisco, San Francisco, CA, USA. [30]Epidemiology, Institute of Public Health Genetics, School of Public Health, University of Washington, Seattle, WA, USA. [31]Internal Medicine, Section on Nephrology, Wake Forest School of Medicine, Winston-Salem, NC, USA. [32]Duke Molecular Physiology Institute, Duke University Medical Center, Durham, NC, USA. [33]Department of Pediatrics, Genomic Outcomes, The Institute for Translational Genomics and Population Sciences, Department of Pediatrics, The Lundquist Institute for Biomedical Innovation at Harbor-UCLA Medical Center, Torrance, CA, USA. [34]Preventive Medicine, Keck School of Medicine, University of Southern California, Los Angeles, CA, USA. [35]Department of Epidemiology, School of Public Health, University of Alabama at Birmingham School of Public Health, Birmingham, AL, USA. [36]Department of Epidemiology, School of Public Health, UTHealth Houston, Houston, TX, USA. [37]Veterans Affairs Boston Healthcare System, Boston, MA, USA. [38]Division of Pulmonary and Critical Care Medicine, Brigham and Women's Hospital, Harvard Medical School, Boston, MA, USA. [39]Department of Epidemiology, Emory University Rollins School of Public Health, Atlanta, GA, USA. [40]Atlanta VA Health Care System, Decatur, GA, USA. [41]Department of Medicine, Division of Endocrinology and Metabolism, Taipei Veterans General Hospital, Taipei, Taiwan, Taiwan. [42]Endocrinology, Ohio State University, Columbus, OH, USA. [43]Department of Medicine, Endocrinology, Johns Hopkins University School of Medicine, Baltimore, MD, USA. [44]Department of Epidemiology, School of Public Health and Tropical Medicine, Tulane University, New Orleans, LA, USA. [45]Department of Pediatrics, Bay Area Pediatrics, Oakland, CA, USA. [46]Epidemiology Program, University of Hawaii Cancer Center, Honolulu, HI, USA. [47]Department of Medicine, Pulmonary, Allergy and Critical Care, University of Alabama at Birmingham, Birmingham, AL, USA. [48]Department of Epidemiology, Human Genetics and Environmental Sciences, School of Public Health, The University of Texas Health Science Center at Houston, Houston, TX, USA. [49]Naseri & Associates Public Health Consultancy Firm and Family Health Clinic, Apia, Samoa. [50]International Health Institute, Brown University, Providence, RI, USA. [51]Department of Medicine, Program for Personalized and Genomic Medicine, University of Maryland, Baltimore, MD, USA. [52]Department of Medicine, Brigham and Women's Hospital, Harvard Medical School, Boston, MA, USA. [53]Division of Epidemiology and Community Health, School of Public Health, University of Minnesota, Minneapolis, MN, USA. [54]Department of Medicine, School of Medicine, University of Maryland, Baltimore, MD, USA. [55]Center for Biostatistics and Data Science, Washington University in St. Louis, St. Louis, MO, USA. [56]Department of Medicine, Rheumatology, National Jewish Health, Denver, CO, USA. [57]Lutia i Puava ae Mapu i Fagalele, Apia, Samoa. [58]Medicine, Pharmacology, and Biomedical Informatics, Clinical Pharmacology and Cardiovascular Medicine, Vanderbilt University Medical Center, Nashville, TN, USA. [59]Centro de Neumologia Pediatrica, San Juan, PR, USA. [60]Survey Research Center, Institute for Social Research, University of Michigan, Ann Arbor, MI, USA. [61]Department of Biostatistics, University of Alabama at Birmingham School of Public Health, Birmingham, AL, USA. [62]Department of Medicine, School of Medicine, Boston University, Boston, MA, USA. [63]Department of Human Genetics, School of Public Health, University of Pittsburgh, Pittsburgh, PA, USA. [64]Department of Biostatistics and Health Data Science, School of Public Health, University of Pittsburgh, Pittsburgh, PA, USA. [65]Department of Epidemiology, Indiana University, Indianapolis, IN, USA. [66]Department of Medicine, Indiana University, Indianapolis, IN, USA. [67]Diabaetes Translational Research Center, Indiana University, Indianapolis, IN, USA. [68]Department of Medicine, Emory University School of Medicine, Atlanta, GA, USA. [69]Department of Medicine, Cardiovascular Medicine, Vanderbilt University Medical Center, Nashville, TN, USA. [70]Department of Biostatistics, Department of Psychiatry, University of Michigan, Ann Arbor, MI, USA. [71]Department of Epidemiology, Arnold School of Public Health, University of South Carolina, Columbia, SC, USA. [72]Department of Human Genetics and South Texas Diabetes and Obesity Institute, School of Medicine, University of Texas Rio Grande Valley, Brownsville, TX, USA. [73]Bioengineering and Therapeutic Sciences and Medicine, Lung Biology Center, University of California, San Francisco, San Francisco, CA, USA. [74]Department of Medicine, University of Mississippi Medical Center, Jackson, MS, USA. [75]Division of Preventive Medicine, Brigham and Women's Hospital, Boston, MA, USA. [76]Harvard Medical School, Boston, MA, USA. [77]Department of Medical Genetics, Genomic Outcomes, Lundquist Institute for Biomedical Innovation at Harbor-UCLA Medical Center, Torrance, CA, USA. [78]Brown Foundation Institute of Molecular Medicine, McGovern Medical School, University of Texas Health Science Center at Houston, Houston, TX, USA. [79]Department of Medicine, School of Medicine, University of Illinois at Chicago, Chicago, IL, USA. [80]Department of Epidemiology, University of Washington, Seattle, WA, USA. [81]Northwestern University, Chicago, IL, USA. [82]Department of Epidemiology, University of Alabama at Birmingham School of Public Health, Birmingham, AL, USA. [83]Department of Medicine, Division of Endocrinology, Diabetes and Nutrition, University of Maryland, Baltimore, MD, USA. [84]Department of Medicine, School of Medicine, University of Pittsburgh, Pittsburgh, PA, USA. [85]Department of Health Systems and Population Health, University of Washington, Seattle, WA, USA. [86]Department of Genetics, University of North Carolina at Chapel Hill, Chapel Hill, NC, USA. [87]Center for Public Health Genomics, University of Virginia, Charlottesville, VA, USA. [88]Department of Epidemiology, School of Public Health, University of Washington, Seattle, WA, USA. [89]Kaiser Permanente Washington Health Research Institute, Kaiser Permanente Washington, Seattle, WA, USA. [90]Seattle Epidemiologic Research and Information Center, Office of Research and Development, Department of Veterans Affairs, Seattle, WA, USA. [91]Department of Medicine, Division of Hematology, Duke University School of Medical, Durham, NC, USA. [92]Department of Medicine, Channing Division of Network Medicine, Harvard Medical School, Boston, MA, USA. [93]Framingham Heart Study, School of Medicine, Boston University Chobanian & Avedisian School of Medicine, Boston, MA, USA. [94]Department of Statistics, Harvard University, Cambridge, MA, USA. [95]Novo Nordisk Foundation Center for Basic Metabolic Research, Faculty of Health and Medical Sciences, University of Copenhagen, Copenhagen, Denmark. [96]These authors contributed equally: Xinruo Zhang, Jennifer A. Brody, Mariaelisa Graff, Heather M. Highland, Nathalie Chami. [98]These authors jointly supervised this work: L. Adrienne Cupples, Leslie A. Lange, Ching-Ti Liu, Ruth J. Loos, Kari E. North, Anne E. Justice. [99]Deceased: Diane M. Becker, Yen-Feng Chiu, Rebecca D. Jackson, L. Adrienne Cupples.  *A list of authors and their affiliations appears at the end of the paper. ✉e-mail: xinruo@email.unc.edu; aejustice1@geisinger.edu

## NHLBI Trans-Omics for Precision Medicine (TOPMed) Consortium

Xinruo Zhang ⓘ [1,96]✉, Jennifer A. Brody ⓘ [2,96], Mariaelisa Graff ⓘ [1,96], Heather M. Highland ⓘ [1,96], Nathalie Chami ⓘ [3,4,96], Hanfei Xu[5], Zhe Wang ⓘ [3], Xihao Li ⓘ [8,9,10], Zilin Li[11,12], Matthew A. Allison[13], Diane M. Becker[14,98], Lawrence F. Bielak ⓘ [15], Joshua C. Bis ⓘ [2], Meher Preethi Boorgula[16], Donald W. Bowden[17], Jai G. Broome ⓘ [18,19], Erin J. Buth[18], Christopher S. Carlson[20], Kyong-Mi Chang ⓘ [21,22], Sameer Chavan ⓘ [16], Yen-Feng Chiu ⓘ [23,98], Lee-Ming Chuang ⓘ [24], Matthew P. Conomos ⓘ [18], Dawn L. DeMeo[25], Mengmeng Du[26], Ravindranath Duggirala[27,28], Celeste Eng[29], Alison E. Fohner ⓘ [30], Barry I. Freedman[31], Melanie E. Garrett ⓘ [32], Xiuqing Guo ⓘ [33], Chris Haiman[34],

Benjamin D. Heavner [18], Bertha Hidalgo [35], James E. Hixson[36], Brian D. Hobbs[25,38], Donglei Hu [29], Chii-Min Hwu [41], Rebecca D. Jackson[42,98], Deepti Jain[18], Rita R. Kalyani[43], Sharon L. R. Kardia[15], Tanika N. Kelly[44], Ethan M. Lange[6], Michael LeNoir[45], Changwei Li [44], Loic Le Marchand[46], Merry-Lynn N. McDonald [47], Caitlin P. McHugh[18], Alanna C. Morrison [48], Take Naseri[49,50], Jeffrey O'Connell[51], Nicholette D. Palmer [17], James S. Pankow [53], James A. Perry [54], Ulrike Peters [20], D. C. Rao[55], Elizabeth A. Regan[56], Sefuiva M. Reupena[57], Dan M. Roden [58], Jose Rodriguez-Santana[59], Colleen M. Sitlani[2], Jennifer A. Smith [15,60], Hemant K. Tiwari[61], Ramachandran S. Vasan [62], Daniel E. Weeks [63,64], Jennifer Wessel [65,66,67], Kerri L. Wiggins [2], Lynne R. Wilkens[46], Lisa R. Yanek [14], Zachary T. Yoneda[69], Wei Zhao [15,60], Sebastian Zöllner [70], Donna K. Arnett[71], Allison E. Ashley-Koch [32], Kathleen C. Barnes [16], John Blangero [72], Eric Boerwinkle[48], Esteban G. Burchard [73], April P. Carson [74], Daniel I. Chasman [75,76], Yii-Der Ida Chen[77], Joanne E. Curran [72], Myriam Fornage [48,78], Victor R. Gordeuk [79], Jiang He [44], Susan R. Heckbert [2,80], Lifang Hou[81], Marguerite R. Irvin[82], Charles Kooperberg [20], Ryan L. Minster [63], Braxton D. Mitchell [83], Mehdi Nouraie[84], Bruce M. Psaty [2,80,85], Laura M. Raffield[86], Alexander P. Reiner [80], Stephen S. Rich [87], Jerome I. Rotter [33], M. Benjamin Shoemaker[69], Nicholas L. Smith [88,89,90], Kent D. Taylor[33], Marilyn J. Telen [91], Scott T. Weiss [92], Yingze Zhang [84], Nancy Heard-Costa [93], Xihong Lin [8,94], L. Adrienne Cupples [5,97,98], Leslie A. Lange[6,97], Ching-Ti Liu [5,97], Ruth J. F. Loos [3,4,95,97], Kari E. North [1,97] & Anne E. Justice [7,97] ✉

A full list of members and their affiliations appears in the Supplementary Information.

