## [Peer Review file · Nature Communications]

Whole Genome Sequencing Analysis of Body Mass Index Identifies Novel African Ancestry-Specific Risk Allele

Corresponding Author: Dr Anne Justice

Version 0:

Reviewer comments:

Reviewer #1

(Remarks to the Author)

This study is a trans-ancestry genome-wide association study of body mass index, using whole genome sequencing data from TopMed. It identifies 18 signals, of which 1 is a novel locus. The study utilizes whole genome sequencing from ~88,000 participants from TopMed and so represents a substantial undertaking to evaluate the genetic etiology of BMI. Larger genome-wide association studies and exome sequencing studies have identified many variants for BMI, limiting the space for common loci, but this study represents an opportunity for a deep dive into genetic susceptibility. The study includes multiple populations, but some of the diversity and perhaps insight into population differences is lost in the analysis, which amalgamates all populations together in a single analysis with separate analyses for European and African populations only.

Specific comments:

1. The authors analyze all populations together in a single analysis, adjusting for ancestry and principal components. However, if there are true differences in the effects across populations for a particular variant (e.g., perhaps due to environmental differences), is modeling this heteroscedasticity in the LMM the best approach? Were similar results observed by conducting analyses separately by ancestry and then combining them in a meta-analysis using MR-MEGA or some other approach? For the conditional analysis, it is not clear how well simply conditioning on the top SNP works when the populations are aggregated together, given the different LD structures across populations and potentially different effects of the variant on risk. Are the results the same if the conditional analysis is run separately for each population (e.g., each population conditioned on the same top SNP from the trans-ancestry analysis) and then the population-specific SNP results (after conditional analysis) are meta-analyzed together?
2. The authors provide the population-specific allele frequencies for the variants that reach genome-wide significance, but not the population-specific betas and percent variance explained. Is there evidence of heterogeneity across populations for these loci/signals? Do the betas and % variance explained differ across populations?
3. The replication for the MTMR3 locus was done in cohorts with similar populations, but the locus appears specific to African ancestry. What were the replication results show when limited to African ancestry in these cohorts? Is there any evidence that the variant affects gene expression or protein levels? The authors do not believe that the SNP reported is functional, but given that they have whole genome sequence data available, is it possible to look at haplotypes and/or additional functional annotations beyond what is VEP (which does not appear cell-type specific) that might provide more information. Is the MTMR3 variant independent of the previously reported variant for weight and fat-free mass?
4. Speliotes et al (2010) reported a secondary signal at MC4R. Is the secondary signal reported here independent of the previously reported secondary signal or just a different secondary signal perhaps due to differences in LD? Is it truly new? The authors state that it is not robust to conditional analyses using previously reported variants, which suggests that it does not provide any information beyond the 2 SNPs reported previous in Speliotes or elsewhere. Sup data 9 and 10 should indicate exactly which SNP(s) were used for the conditional analysis. Also, why are some of the SNPs in Sup data 9 different from Table 1? Please clarify.
5. There have been a number of SNPs and genes from exome sequencing reported for BMI previously (e.g., Akbari, Science 2021). Were these SNP and gene-based associations from Akbari et al replicated in this study? The authors did aggregate testing and said that they replicated the gene-based MC4R finding, but what about the findings from Akbari? Given the number of different populations in the dataset, evaluating these previously report SNPs across populations seems like a missed opportunity.
6. Fine mapping was done using the multi-ancestry summary statistics with PAINTOR, but such an analysis does not fully capture the LD structure of the different populations included and highly dependent on the functional annotation. The

functional annotation based on GeneHancer does not appear to be tissue or cell-type specific, which may limit its ability to accurately fine-map the loci. Were other annotations considered and tested, such as regulatory elements from neuronal tissue?

7. Not much detail is given regarding the assessment of BMI. Were weight and height measured in the participants or was the data obtained through self-report?

8. Minor clarification on methods (page 13): The authors state that LDlink was used to calculate pairwise LD between potentially independent signals in known loci and produce LD heatmaps using 1K data. In other parts of the paper, TopMed was used to estimate LD. Why wasn't the data from TopMed used to estimate pairwise LD and produce heatmaps?

Reviewer #2

(Remarks to the Author)

This manuscript describes a cross-ancestry genetic analysis of BMI focusing on non-European cohorts. For the most part it identifies known BMI associations, and it also identifies a potentially novel association with the variant rs111490516 in MTMR3, which is fixed in individuals with European genetic similarity and has various frequencies in other populations.

It is nice to be able to show that analyzing beyond European genetic similarity can identify novel results. I am still a little torn about whether or not the MTMR3 hit is real, but overall the authors have done a good job of investigating from as many avenues as possible to try to show that this result holds up.

I would advise the authors to use the phrasing "genetic similarity" in their paper following the latest recommendations <https://nap.nationalacademies.org/catalog/26902/using-population-descriptors-in-genetics-and-genomics-research-a-new>.

While it is becoming more common to analyze data across populations, especially in collapsing analyses of rare variants, I'm not sure the field is convinced that these types of analyses are always appropriate when analyzing individual variants. It is still typical to see analyses performed on each ancestry separately, and I was glad to see the European and African separate analyses here to shore up the results. I would however like to see more citations and explanations of why it's ok to analyze all the ancestries together and what reassurances there are that biases are not occurring. Also, in the European and African-specific analyses, I was not sure if new PCs specific to those genetic similarity groups had been generated and used for the analysis, or if the PCs that were generated for the whole group were what was used still. In a group-specific analysis, PCs generated specifically for that genetic similarity group should be used.

However, when discussing the group-specific analyses, the authors indicate that rs111490516 was not the genome-wide significant hit in MTMR3 in the African analysis, and a different variant rs73396827 is associated instead. This needs more exploration. What was the p-value for rs111490516 in this analysis? What was the p-value or beta for rs111490516 and rs73396827 in the other genetic similarity groups included in the cross-ancestry analysis? There is a history in human genetics of putting too much emphasis on replication that is not actually replication, i.e., seeing a signal with another variant in the gene instead of the original one and not properly correcting for how many variants in the gene were tested. Here it sounds like the variants are in very high LD in the African genetic similarity group, so this shouldn't be an issue, but it is helpful to explain further what is going on.

Furthermore for the MTMR3 variant, in table s7, it is shown that the beta is lower and the pval nowhere near significant in the individual replication groups, before the meta analysis. I know there is winners curse, but the similar sample size of MVP at least to the discovery cohort and the substantial difference in beta and pval of 0.04 despite the allele frequency being 3x higher in MVP, and therefore should have better power, is rather concerning. Do the others have a hypothesis for why the signal is so questionable in MVP in particular? Is it because the imputation likely did not go well? Overall, if imputation accuracy is a concern, can the authors use the UKB WGS data instead of imputation to help recover the signal? For me, Table S7 and especially the MVP numbers were the part of the paper that made me the most hesitant about the finding. It may still be a real association, but we need more explanation for what is going on here. The explanation given in the second paragraph of the discussion that the lack of clear replication may be because it's not clear which variant is causal in the region is not very convincing to me.

A small thing, one phrase bothered me: "In the two population-specific analyses, 10 association signals reached genome-wide significance 387 (Supplementary Data 6, Supplementary Figs 4 – 7). Two of these signals were also detected in the 388 multi-population analysis." What are the other 8 hits, are they known hits? Why don't they come up in the cross ancestry analysis? This appears better explained in supplementary data 6 that they are in LD with hits in the main analysis, but in that case this sentence should be rephrased I think.

Reviewer #3

(Remarks to the Author)

Whole genome sequencing analysis of body mass index identifies novel African ancestry-specific risk allele

This manuscript by Zhang, Brody, Graff, Highland, Chami, et al. reports on a study of whole genome sequencing data from among almost 90,000 participants of high ancestral diversity when compared to the vast majority of genetic datasets in the field. The authors analysed these data to systematically investigate genetic associations with body mass index (BMI). Of the index associations they find, one was novel, and common among individuals of African descent. After finemapping, they report two putatively causal variants in POC5 and DMD loci.

The study was generally carefully conducted, and made use of a variety of approaches to interrogate the genetic data. Importantly, the authors went to the trouble of replicating the association found in the discovery sample through a meta-analysis of MEC, MVP, BioMe, UKBB, and REGARDS. I also applaud the authors for reporting the negative result in the rare-variant association testing. The technical execution of the work appears robust, and the study is well-written. However, I do have a few reservations:

Major comments

1) A potential issue of the study is that the title implicitly suggests that the novel association would not have been found without whole genome sequencing data. The authors allude to this in the discussion: “our index SNP [rs111490516] is likely not causal but could be in LD with a causal SNP and also poorly captured in studies relying on imputation”. On its face this is very interesting, and warrants investigation: is the variant poorly imputed because of European focused SNP array design not allowing the variant to be imputed? Is it because we are lacking imputation panels that would allow for better imputation of array data from individuals of African descent? That said, Supplementary Data 7 seems to suggest that current imputation panels do quite a good job of imputing the variant, so do we really require sequencing data to determine the genotype at rs111490516 accurately? It seems that this question could be answered by taking the discovery set (TOPMED), masking the variants not present on a set of SNP arrays (e.g. that used for MVP, given that it displayed the lowest R²/info score), and then determine whether the association signal would have been recapitulated using HRC or 1000G as a reference panel. If the signal could not be recapitulated using this approach, it would lend a huge amount of evidence to one of the central assertions of the paper.

2) As it currently stands, the quality control and analyses in the paper are not reproducible. While I understand that detailed descriptions of the methods cannot be provided in detail in the main text due to length restrictions, they should be present in complete detail in the supplement. For example, what thresholds have been used in HARE? How were PCs defined? How many PCs were used in the analyses? How was sex concordance defined? In pheWAS, what thresholds were used to define ‘African’ ancestry? Etc, etc.

Also, there is no code provided to reproduce the analyses. Can you please link to the code used for the QC and analyses in a publicly available repository (e.g. on github)? Upon clicking the link to find the methods for quality control on the TOPMed website, I get ‘access denied’ and told to email regarding membership. These methods could be extremely valuable to the genetics community, and would have the greatest impact and use if they were not only available to members of TOPMed. Sharing of code and detailed methods would be particularly impactful given the need for careful analysis of data from diverse ancestries.

3) Availability of results. It would be great to see full variant and gene-based association test statistics made available to the community. As it stands, there is currently no data availability statement for the analysis results. Will all summary statistics be made publicly available?

4) The claim that “no previous study has successfully identified BMI risk variants of high confidence at the POC5 and DMD loci” in the discussion appears to be immediately contradicted “these two variants were also considered high-confidence causal variants (PPrs2307111 = 0.96, PPrs1379871 = 0.99) in a recent joint analysis of three biobanks (UKBB, FG, BBJ)”. Is this statement specifically referencing the analysis of non-European populations? Further, the claim that because the variant is at high prevalence across multiple populations implies that it is more likely to be causal seems speculative and should be backed up. If this is what you are claiming, why should we conclude this?

“Notably, unlike in Kanai et al. where the PP appeared to be driven by the Europeans (for rs2307111: PPUKBB = 0.96, PPBBJ = 0.12, PPFPG = 0.01; for rs1379871: PPUKBB = 1.00, PPFPG = 1.00), the effect alleles in our study were observed in high proportions across many non-European population groups”

Minor comments

1) Why do you use 9 PCs in HARE? Looking at the paper, they suggest 30. What metric did you use to determine that 9 was the appropriate cutoff? Scree plot?

2) References are missing for the various annotations used in the rare variant association analyses (LOFTEE, MetaSVM, FATHMM-XF). Can the authors outline why these particular metrics were used in preference to others, particularly given the recent influx of damaging missense prediction metrics?

3) For the rare variant collapsing analysis, it would be interesting to see more of a deep-dive into the different classes of variant and a combined analysis e.g. via a Cauchy combination test, as in other publications using e.g. SAIGE-gene or REGENIE. As it stands, although the results are null, it seems like a missed opportunity for a deep investigation of rare-variant group based testing. Did any other (non-novel) genes pop up in the gene-based analysis, or was it only MC4R?

4) In pheWAS, only removing up to second degree relatives seems permissive (assuming that a mixed model was not used to run the association at the locus). Are the authors sure that the analysis would be robust if they had detected a pheWAS hit? Given the restriction in bioME to samples with at least 20 cases or controls, that would imply that a large number of the phenotypes will have large case-control imbalance. It would be helpful to know what tests were used, and that the

association test statistics were checked for robustness.

5) The analysis would benefit from a more thorough investigation into the functional consequences of the finemapped SNPs. As currently written, it seems like a bit of an afterthought.

6) It would be helpful to know from the MVP investigators why rs111490516 was removed in their quality control pipeline.

7) How do you arrive at 18 variants? Are the remaining two variants over the 16 reported in the single-variant analysis section those that are in tight LD with the index SNPs in the multi-population analysis? If so, should they really count as an additional two associations?

8) PAINTOR now allows multiple causal associations at a locus. Why was a single causal variant assumed in finemapping? If the assumption was relaxed, could that be used to better ascertain the likely causal variant in the locus, and also to check robustness of the implicated variants when assuming a single causal variant? It may also be beneficial to compare and contrast the results with other methods e.g. SuSiEx.

Version 1:

Reviewer comments:

Reviewer #1

(Remarks to the Author)

The authors have addressed my concerns.

Reviewer #2

(Remarks to the Author)

Thank you for addressing many of my concerns. I would still like to see the African-only analysis done with African-specific PCs. Given the higher mean BMI in African ancestry in this study and the higher allele frequency in African ancestry of the novel implicated variant, the best way to make sure that there is no confounding by percent or type of African ancestry in the results is to use African-specific PCs as covariates. It is also field standard to use ancestry-specific PCs when doing an ancestry-specific analysis.

Reviewer #3

(Remarks to the Author)

Please find my responses to each of the numbered rebuttals.

Major comments

- 1)
Thank you for looking into this, but it seems odd to claim that it is not within scope since your title implies that it is through sequencing that you detect the variant and would not have discovered it otherwise.
- 2)
Thank you for providing information to aid in reproducibility of results. However, the link seems to only include the quality control of datasets and variant calling, but not the analyses presented in this paper. Zenodo links to the code used to perform your analyses would go a long way to ensuring reproducibility of the presented results.
- 3)
Thank you.
- 4)
This is now much clearer, thank you.

Minor comments

- 1)
Thank you.
- 2)
Excellent, thank you. Further clarification described in the reviewer response for why LOFTEE and other tools were chosen would be useful to include in article methods section.
- 3)
Thank you. It would be good to gain an understanding for why the remaining hits in Akbari were not found - is it simply a power issue? You mention the increased flexibility of GENESIS allowing for heterogeneity of effect size. It would be helpful to reference this advantage in the methods, and why other approaches were not used. The motivation and backing up of your methods over the meta-analysis as suggested by reviewers 1 and 2 is important and should be discussed in the article (and not relegated to the reviewer rebuttal).
- 4)
- 5) Excellent, thank you.
- 6) Thank you for the clarification.
- 7) Thank you for clarifying.

8) 'As PAINTOR may be sensitive to misspecification of the number of causal variants' seems an understatement. Thank you to the authors for investigating. Your examination of the loci here is helpful and should be included in the supplement of the paper. What is your reference for >25kb LD decays, and what does this mean? $R^2 < X$ for some large proportion Y% of the time? What are X and Y? Without X and Y, the statement doesn't tell us a great deal. What is the LD between your primary signal and secondary signals? Doesn't the secondary signal being >25kb away from your primary signal contradict your LD decay claim?

Version 2:

Reviewer comments:

Reviewer #2

(Remarks to the Author)

the authors have addressed my concerns

Reviewer #3

(Remarks to the Author)

Firstly, regarding the response to reviewer 2, I really don't understand why the results you present (in the table) cannot be included in the manuscript. They back up your use of mega-analyses, which as the reviewer mentions, is not standard practice. The statement "it is not the standard practice within TOPMed to re-calculate PCs in population subsets", seems an odd reason not to include the additional results. In fact, I would encourage the sharing of the entire AFR only GWAS sumstats alongside your primary mega meta-analysis sumstats.

Reviewer #3 (Remarks to the Author): Please find my responses to each of the numbered rebuttals. (Note, we removed previous comments that did not require an additional response for this reviewer.)

Major comments

Reviewer – Round 2 (1): Thank you for looking into this, but it seems odd to claim that it is not within scope since your title implies that it is through sequencing that you detect the variant and would not have discovered it otherwise.

We agree with the reviewer that addressing the disparity between imputation quality and sequencing that leads to either new discoveries, non-replication of previous discoveries, or changes in power is an interesting topic. However, our focus for this paper was to leverage the sequencing for discovery of new variants, understanding that there are multiple reasons as to why new discoveries may be possible. Through whole genome sequencing of a large number of ancestrally diverse individuals, a much larger number of variants are able to be imputed to the truth. However, data use agreements for some studies within TOPMed provide limitations on the use of data, including for imputation. Thus, it would breach our agreements to use these data for those purposes. Further, the largest discovery analysis of BMI to date is heavily biased towards European ancestry individuals and is only imputed to HapMap3 and HRC. Teasing apart the cause of the novel discovery is a very broad topic that could be covered in a separate paper. Indeed, there are publications that are looking at the improvement in imputation with the advent of more diverse reference panels. <https://www.biorxiv.org/content/10.1101/2023.05.22.541241v1.full>.

Thank you. Perhaps mentioning evidence of poorer imputation accuracy in non-European populations is likely to be an issue, even for e.g. the 'global' screening array, means that it makes sense to look in WGS data, which is becoming more and more ubiquitous, acknowledging that you're not actually able to test this assertion in your data - highlighting the data use agreements, in the discussion.

Reviewer – Round 2 (2): Thank you for providing information to aid in reproducibility of results. However, the link seems to only include the quality control of datasets and variant calling, but not the analyses presented in this paper. Zenodo links to the code used to perform your analyses would go a long way to ensuring reproducibility of the presented results.

Response – Round 2: We apologize for the confusion. We understand that there are many links presented. The code for our analysis was carried out using this specific pipeline: https://github.com/AnalysisCommons/genesis_wdl, under the R scripts "genesis_nullmodel.R", "genesis_tests.R", "pipelineFunctions.R", etc.

Thank you. This makes sense, but https://github.com/AnalysisCommons/genesis_wdl, is general use code for pipelining GENESIS, rather than your code to perform call GENESIS in your primary analysis, and all of the subsequent code that you would have used for your subsequent post-GWAS analysis. Please include your code for GWAS (including all options passed in the wdl), and post-GWAS analysis (including all versions of software used, and options/flags used therein, and all scripts), with Zenodo.

Minor comments Reviewer – Round 2 (2): Excellent, thank you. Further clarification described in the reviewer response for why LOFTEE and other tools were chosen would be useful to include in article methods section.

Response – Round 2: Thank you for the recommendation. We have now edited our previous response under 'Rare Variant Aggregate Association Analysis' section, as shown below.

Rare Variant Aggregate Association Analysis

Rare variants with a MAF $\leq 1\%$ were tested in aggregate by gene unit across studies in the multipopulation analysis. Variants were grouped into gene units in reference to GENCODE v28, including both coding variants and variants falling within gene-associated non-coding elements. Coding variants included high-confidence loss of function variants annotated by LOFTEE (Ensembl VEP LoF = HC), missense variants (MetaSVM score > 0) and in-frame insertion/deletions or synonymous variants (FATHMM-XF coding score > 0.5). In addition to coding variants, we included variants falling within the promoter of each transcript tested. Promoter regions were defined as falling in the 5 kb region 5' of the transcript and also overlaying a FANTOM5 Cage Peak. In order to identify regulatory elements likely to be acting through the tested gene, we leveraged the GeneHancer database. GeneHancer identifies enhancer regions and associates them with the specific genes they are likely to regulate, allowing us to aggregate regulatory regions by the likely target gene. GeneHancer regions were limited to the top 50% scored regions and variants falling in these regulatory elements were further filtered to those most likely to have a functional impact (FATHMM-MKL noncoding score > 0.75). Variants aggregated to gene units were tested using variant-set mixed model association tests (SMMAT). Variants were weighted inversely to their MAF using a beta distribution density function with parameters 1 and 25. Genes were considered significantly associated after Bonferroni correction for the number of genes analyzed ($P < 5 \times 10^{-7}$). These annotations were selected by the TOPMed Data Coordinating Center (DCC) as part of the centralized and harmonized annotations, and were chosen to focus on annotation that was previously shown to have high agreement with other annotation resources and high prediction accuracy.

Thank you. Can you provide some references for your assertion that "chosen to focus on annotation that was previously shown to have high agreement with other annotation resources and high prediction accuracy."

Reviewer – Round 2 (3): Thank you. It would be good to gain an understanding for why the remaining hits in Akbari were not found - is it simply a power issue? You mention the increased flexibility of GENESIS allowing for heterogeneity of effect size. It would be helpful to reference this advantage in the methods, and why other approaches were not used. The motivation and backing up of your methods over the meta-analysis as suggested by reviewers 1 and 2 is important and should be discussed in the article (and not relegated to the reviewer rebuttal).

Response – Round 2: Replication of gene-based findings is extremely nuanced. Akbari et al. had a much larger samples (~7 times larger), including multiple studies with related individuals. In addition to sample size, differences in alleles harbored by individuals in a given dataset, grouping of variants, and rare variant test all can have large impact on significance of gene-based test. Failure to replicate gene-based signals must be interpreted with extreme caution. In addition, the results published by Akbari et al. presented findings at multiple genes, but they only attempted to replicate and validate one gene-based association, therefore there is still question about the robustness of the remaining findings and whether we should expect to be able to replicate them. Therefore, it is likely a combination of power due to differences in sample size, underlying variant selection, and difference in methods for gene-based analysis. We have added further explanation about the possible lack of replication in the Discussion section. Also, we have now provided further clarification on GENESIS for our analysis plan:

“Common Variant Association Analysis We performed multi-population WGS association analysis of BMI using GENESIS on the Analysis Commons (<http://analysiscommons.com>) computation platform. GENESIS was chosen due to its analytical flexibility in relationship to allowing for heterogeneity of effect by population group, an option well-suited to the demographic and genetic background of our study population. Analyses were performed using linear mixed models (LMM). To improve power and control for false positives with a non-normal phenotype distribution, we implemented a fully adjusted two-stage procedure for ranknormalization when fitting the null model¹⁵...

...Due to the large number of variants tested ($N = 90,142,062$) in the multi-population analysis, we adopted a significance threshold of 5×10^{-9} as has been used previously¹⁹. This approach maximizes participant inclusion, thereby maximizing the sample size in our discovery cohort to increase statistical power and avoiding the misinterpretation of group-specific effects in underpowered strata. By including ancestrally diverse populations, we leverage differences in LD across populations, which has been shown to help identify novel loci, narrow down causal variants, and improve the variance explained in models (PMIDs: 26748518, 31217584, 36119389). For quantitative traits like BMI, multi-population analyses that account for heteroscedasticity of genetic effects across population groups have proven effective in increasing study power and reducing genomic inflation (PMIDs: 26748518, 31217584). Lastly, and most importantly, a pooled approach aids in decreasing health disparities by identifying loci that generalize across populations.

Additionally, group-specific analyses were conducted in the two largest population groups, European and African, to determine whether a particular group with large sample size is driving the observed association signal.

“In addition to our novel findings, 17 of the 18 identified variants reside in previously reported BMI-associated loci, highlighting the generalizability of the genes underlying BMI across populations, including SEC16B, TMEM18, ETV5, GNPDA2, BDNF, and MC4R. Three of the loci harbor genes implicated in severe and early-onset obesity – ADCY3, BDNF, and MC4R. We also consistently identified multiple association signals of high effect in MC4R, which is a well-established monogenic obesity gene, through our discovery analysis, internal conditional analysis, and rare variant aggregate analysis. Despite not identifying novel SNPs in MC4R that are independent of known BMI-associated SNPs, we replicated a secondary signal in this gene, rs2229616, a rare missense variant previously reported in individuals of European ancestry by Speliotes et al. In addition to MC4R, three other genes – ROBO1, GPR151, and ANO4 – of the exome-wide significant genes identified in Akbari et al. showed nominal significance in our rare variant aggregate analysis. This lends further support to the generalizability of these genes across populations, given that 85% were of European ancestry and 15% of admixed American ancestry in Akbari et al., compared to our more diverse cohort with 49% European and 51% other

populations. We did not replicate 12 of the gene-based findings from Akbari et al. The previous study had nearly sevenfold larger sample size compared to the current study, included related individuals increasing the opportunity for multiple copies of rare variants, and implemented alternative methods for gene-based analysis and variant binning. Therefore, it is likely a combination of power due to differences in sample size, underlying variant selection, difference in methods for gene-based analysis, and potential winner's curse that contributed to the lack of validation of their findings in the current study."

Thank you.

Reviewer – Round 2 (8) Reviewer – As PAINTOR may be sensitive to misspecification of the number of causal variants' seems an understatement. Thank you to the authors for investigating. Your examination of the loci here is helpful and should be included in the supplement of the paper.

What is your reference for $>25\text{kb}$ LD decays, and what does this mean? $R^2 < X$ for some large proportion $Y\%$ of the time? What are X and Y ? Without X and Y , the statement doesn't tell us a great deal. What is the LD between your primary signal and secondary signals? Doesn't the secondary signal being $>25\text{kb}$ away from your primary signal contradict your LD decay claim?

We based our window size on the recommendations of Greenbaum and Deng 2017 (PMID: 28425624) that suggest a window of 100 kb based on the LD decay $>25\text{ kb}$. The observation of decay $>25\text{ kb}$ is based on evidence of the portability of LD blocks up to 20 kb windows across multiple HapMap populations ($>80\%$ of haplotype blocks less than this distance) with a decrease in portability when extending this window, among other analyses that point to decreased LD decay ($R^2 < 0.5$) at greater distances identified by Conrad DF, et al. 2006 (PMID: 17057719). Given that the secondary signal is greater than 25 kb from our primary signal, this supports the conclusion that these are independent loci, given that we expect increased independence (lower LD) at $>25\text{ kb}$ distances. However, we do acknowledge that limiting the window size can cause us to potentially miss causal variants that may be greater than 100 kb from our lead variant. We have modified our fine-mapping methods section to provide a reference for the window chosen and acknowledge this limitation.

We appreciate that the reviewer found our additional PAINTOR analyses helpful; however, given that we are not confident in the results of PAINTOR assuming more than one causal variant for the reasons previously noted, we have chosen to leave these analyses out of the main paper and reserve this to the response to reviewers that will be available alongside the paper. We have edited the methods to clarify further why only one causal variant was assumed:

"We restricted this analysis to variants located within $\pm 100\text{ kb}$ of the locus index variants. We calculated LD using our analysis subset of the TOPMed data. As PAINTOR may be sensitive to the misspecification of the number of causal variants, limiting our ability to interpret findings in the absence of evidence for more than one signal, we assumed one causal variant per locus, unless evidence of independent secondary signals within the 100 kb window was identified following conditional analysis, in which case we allowed for additional causal variants per locus. Restriction to a $100\text{ kb} \pm$ window was applied due to the computational burden of WGS data and because, for most loci, LD decays at $> 25\text{ kb}$ (PMIDs: 28425624, 17057719). While we extend our fine-mapping locus to 100 kb to allow for potential LD beyond 25 kb, one limitation of this approach is that we may still miss potentially causal variants that are $>100\text{ kb}$ from our index SNP, including secondary signals."

I disagree with the exclusion of the comparative PAINTOR analysis assuming multiple causal variants. It is useful given the statement in PMID: 26189819 "In general, we find that trans-ethnic fine-mapping strategies that assume a single causal variant are less optimal than those that allow for multiple causal variants", from the PAINTOR authors. Furthermore, reference 64 seems to suggest that PAINTOR can often return unreasonable results even for a single causal variant, which would seem to suggest that other finemapping approaches should be considered for consistency. Also, both advocate for the use of GWAS within population specific LD, and cross-ancestry finemapping. Stating this in the discussion to improve finemapping would be advisable.

Response to reviewer comments: Whole Genome Sequencing Analysis of Body Mass Index identifies Novel African Ancestry-Specific Risk Allele

We would like to thank all of the reviewers for their thorough and insightful review. We were happy to see that the reviewers point to our study as “a substantial undertaking to evaluate the genetic etiology of BMI” and appreciate the multi-population approach and our efforts to present robust findings. We found the reviewers’ comments incredibly helpful and feel they improved the manuscript, for which we are grateful.

We have attached a point-by-point response to the reviewers’ recommendations (see below) and have made the required editorial changes. Our responses are shown in blue, with edits to the paper noted in the response and highlighted in red in the revised manuscript. We again thank you for your time and thoughtful consideration.

Point-by-point Response to Reviewer Comments

Reviewer #1 (Remarks to the Author):

This study is a trans-ancestry genome-wide association study of body mass index, using whole genome sequencing data from TopMed. It identifies 18 signals, of which 1 is a novel locus. The study utilizes whole genome sequencing from ~88,000 participants from TopMed and so represents a substantial undertaking to evaluate the genetic etiology of BMI. Larger genome-wide association studies and exome sequencing studies have identified many variants for BMI, limiting the space for common loci, but this study represents an opportunity for a deep dive into genetic susceptibility. The study includes multiple populations, but some of the diversity and perhaps insight into population differences is lost in the analysis, which amalgamates all populations together in a single analysis with separate analyses for European and African populations only.

- We are grateful for the time and effort the reviewer has devoted to evaluating our manuscript. We are grateful for your acknowledgment of the novelty of our findings, including the identification of a new locus, and the substantial effort involved in analyzing data from over 88,000 participants. Your constructive feedback has provided us with a clear direction for refining our analysis and enhancing the overall quality of our work. Thank you for contributing significantly to the improvement of our study.

Specific comments:

1. The authors analyze all populations together in a single analysis, adjusting for ancestry and principal components. However, if there are true differences in the effects across populations for a particular variant (e.g., perhaps due to environmental differences), is modeling this heteroscedasticity in the LMM the best approach? Were similar results observed by conducting analyses separately by ancestry and then combining them in a meta-analysis using MR-MEGA or some other approach? For the conditional analysis, it is not clear how well simply conditioning on the top SNP works when the populations are aggregated together, given the different LD structures across populations and potentially different effects of the variant on risk. Are the results the same if the conditional analysis is run separately for each population (e.g., each population conditioned on the same top SNP from the trans-ancestry analysis) and then the population-specific SNP results (after conditional analysis) are meta-analyzed together?

- We appreciate the reviewer's interest in population-specific loci and/or genetic effects and the potential influence of LD structure on the fine-mapping, as this is of much interest to our group as well. We chose our current approach for a few reasons. First, as the reviewer points out, our total sample size is much smaller than other ongoing efforts using array-based data sets, thus a pooled analysis that maximizes sample size would also maximize our ability to detect novel loci. Also, we are interested in identifying novel loci that generalize across U.S. populations. Work by our group and others in the PAGE Study and TOPMed Program (PMIDs: 31217584, 36119389) show, including ancestrally diverse samples, even a small proportion, enables the identification of novel disease susceptibility loci and assists in narrowing in on causal variants by leveraging differences in LD across ancestries, and importantly, aids in decreasing health disparities. Also, ours and other previous work in ancestrally diverse samples revealed that the addition of population variables to analytical models improved the variance explained in association analyses (PMIDs: 26748518, 31217584, 36119389). For quantitative traits, like BMI, multi-population analyses allowing for heteroscedasticity of genetic effects across population groups has been effective in increasing power in studies performed across groups and reducing genomic inflation (PMID: 26748518, 31217584). Last, we chose to define our population groups and conduct pooled analyses as per the recommendations of the TOPMed Program to maximize the inclusion of all participants, improve control of confounding by population stratification, utilize harmonized population groups, and avoid the mis-interpretation of group-specific effects in under-powered strata (PMID: 36119389). Overall, papers showing that across population analyses enabled by meta-analysis are more powerful and allowing for heterogeneity across groups will allow for greater discovery of loci generalizable across populations.

- Our results, for which we did not identify any additional loci in the European and African group, support the previous findings above. We did not pursue any further stratified analyses for these reasons. Indeed, a post-hoc power analysis shows that we would be greatly underpowered for further stratified analyses using the TOPMed recommended population groups (**Response Table 1**), even assuming common minor allele frequencies (MAF) and large effects like observed at our novel locus. Last, future aims of the TOPMed Anthropometry working group include additional analyses in upcoming data freezes where sample sizes may enable expansion of population specific group analyses beyond the two included here while still aligning with TOPMed recommendations on ancestry/race/ethnicity. For example, for population groups with high burden of obesity (e.g. Samoan), there are parallel efforts that focus solely on these population groups and in collaboration with outside collaborators that might improve power and enable population-specific analyses.

**Response Table 1.** Post-hoc power analysis for population group stratified analyses. Power analyses are based on maximum sample size per population group; a large effect size observed in the pooled analysis (0.0783) for the novel locus near *MTMR3;* allele frequencies of 0.005, 0.01, 0.05, 0.15, 0.30; P<5E-9; normalized values (mean =0, SD =1), and an additive genetic model. Analyses carried out in Quanto v1.2.

| **Population Group** | **N** | **0.5% MAF** | **1% MAF** | **5% MAF** | **15% MAF** | **30% MAF** | **~Min MAF for 80% Power** |
|---|---|---|---|---|---|---|---|
| Pooled | 88,873 | 0.0002 | 0.0052 | 0.9113 | 0.9999 | 0.9999 | 4.25% |
| African | 22,488 | 0.0000 | 0.0000 | 0.0130 | 0.5337 | 0.9614 | 20.5% |
| Amish | 1,106 | 0.0000 | 0.0000 | 0.0000 | 0.0000 | 0.0000 | >30% |
| Asian | 1,241 | 0.0000 | 0.0000 | 0.0000 | 0.0000 | 0.0000 | >30% |
| Barbadian | 248 | 0.0000 | 0.0000 | 0.0000 | 0.0000 | 0.0000 | >30% |

Central American	776	0.0000	0.0000	0.0000	0.0000	0.0000	>30%
Costa Rican	341	0.0000	0.0000	0.0000	0.0000	0.0000	>30%
Cuban	2,128	0.0000	0.0000	0.0000	0.0000	0.0002	>30%
Dominican	2,046	0.0000	0.0000	0.0000	0.0000	0.0002	>30%
European	43,434	0.0000	0.0002	0.2070	0.9917	0.9999	9.25%
Han Chinese	1,787	0.0000	0.0000	0.0000	0.0000	0.0001	>30%
Mexican	4,265	0.0000	0.0000	0.0000	0.0005	0.0057	>30%
Puerto Rican	4,991	0.0000	0.0000	0.0000	0.0011	0.0119	>30%
Samoan	1,274	0.0000	0.0000	0.0000	0.0000	0.0000	>30%
South American	695	0.0000	0.0000	0.0000	0.0000	0.0002	>30%
Taiwanese	2,053	0.0000	0.0000	0.0000	0.0000	0.0002	>30%

2. The authors provide the population-specific allele frequencies for the variants that reach genome-wide significance, but not the population-specific betas and percent variance explained. Is there evidence of heterogeneity across populations for these loci/signals? Do the betas and % variance explained differ across populations?

- Please see our response for 1 above addressing why population-specific analyses were not provided. Additionally, we note that large differences in sample size and trait distribution (as seen here) across strata will often lead to misleading detection of significant heterogeneity (see discussions in PMID: 33400613, 26023932, 25880989), even when normalized phenotypes are used. Further, percent variance explained (PVE) is influenced by MAF when effect size estimates are uniform across groups, which can also lead to a misleading interpretation of heterogeneity across population groups in PVE (for discussion, see: PMID 29547618). For these reasons and those mentioned above, we do not provide population group stratified results for our main findings.

3. The replication for the *MTMR3* locus was done in cohorts with similar populations, but the locus appears specific to African ancestry. What were the replication results show when limited to African ancestry in these cohorts? Is there any evidence that the variant affects gene expression or protein levels? The authors do not believe that the SNP reported is functional, but given that they have whole genome sequence data available, is it possible to look at haplotypes and/or additional functional annotations beyond what is VEP (which does not appear cell-type specific) that might provide more information. Is the *MTMR3* variant independent of the previously reported variant for weight and fat-free mass?

- We apologize for any confusion in the presentation of our replication analysis. The replication of the *MTMR3* locus was carried out in participants that self-identified as Africans, African Americans, and Black or a combination of self-report plus genetic similarity to African populations (i.e. self-report plus HARE analysis) across each replication cohort. We have added a column to Supplementary Data 7 to indicate how participants were identified in each study (e.g. genetic similarity and/or self-report).

Additionally, to improve confidence in our replication analysis, we further expanded our replication cohorts from five to six, by including the MyCode Community Health Initiative study, increasing the total

replication sample size from 110,589 to up to 113,802 (Supplementary Data 7). In MyCode, rs111490516 showed a consistent magnitude and direction of effect as the other replication cohorts, but did not reach significance on its own. The updated replication meta-analysis resulted in a combined $P=1.92E-4$ (compared to $P=4.76E-4$ in our previous submission). As requested by Reviewer #2 (details provided later), we also performed replication on rs73396827, the SNP in the *MTMR3* locus that showed genome-wide significance in the African-specific discovery analysis.

To clarify, we have revised the Replication Analysis Methods as follows:

“For the novel single-variant associations identified in the *MTMR3* locus from our discovery analyses that is largely driven by the African population group, we requested replication specifically in Black, African, and African American participants from six independent cohorts (N total = up to 113,802): BioMe BioBank Program, Million Veteran Program (MVP), Multiethnic Cohort (MEC), MyCode Community Health Initiative Study (MyCode), REasons for Geographic And Racial Differences in Stroke (REGARDS) study, and UK Biobank (UKBB).”

Additionally, the Replication section of the Results were modified as follows:

“The replication sample sizes ranged from 3,213 in MyCode to 80,730 in MVP (Supplementary Data 7). In the six replication studies of Black, African, and African American participants, rs111490516 and rs73396827 in *MTMR3* displayed high LD ($R^2 = 1.00$) (Supplementary Data 7). Their MAFs are nearly identical across studies and ranged from 11% to 13%, aligning with the 13% observed in our African and Barbadian groups (Supplementary Data 7). We replicated both variants, demonstrating directionally consistent associations with BMI across the replication studies. In MVP and REGARDS, particularly, both variants reached statistical significance at $P < 0.05$. We observed a 68% reduction in the estimated effects when meta-analyzing across replication studies (rs111490516: $\beta = 0.025$, $SE = 0.007$, $P = 1.92 \times 10^{-4}$, $MAF = 12\%$; rs73396827: $\beta = 0.026$, $SE = 0.007$, $P = 1.50 \times 10^{-4}$, $MAF = 12\%$), compared to the discovery analysis (Figure 3, Supplementary Data 7, Supplementary Figure 8). In the meta-analysis of up to 202,675 individuals from both discovery and replication studies, rs111490516 and rs73396827 both had an estimated effect of 0.36 with a SE of 0.006 and P -values of 2.40×10^{-9} and 1.60×10^{-9} , respectively (Figure 3, Supplementary Data 7).”

- Indeed, as the reviewer suggested, we are not convinced that our top associated SNP in the *MTMR3* locus is the causal SNP. WGS has the advantages over imputed GWAS data by providing high confidence of associations due to direct genotyping and dense coverage of all available variation in a region. Yet, these associations are still subject to some of the same limitations, especially for common variation and common traits, of being subject to linkage disequilibrium and power (sample size and MAF). To provide additional insight into which variants and genes may be implicated in this association region, we have expanded the annotation of all high LD variants to include lookups in FORGEdb, which has tissue-specific information and provides useful summary of regulatory evidence. Also, as the FORGEdb annotations pointed to multiple tissues, we downloaded potential cCRE annotations for the region from ENCODE for three tissue categories implicated in the FORGEdb annotations, blood, brain, and liver and provide regional association plots for visual inspection.

We provide the following additional section to the Methods.

“Functional Annotation

To gain a better understanding of the potential functional consequence of the *MTMR3* locus, we used Ensembl Variant Effect Predictor (VEP) and FORGEdb to annotate all variants in high LD with our top SNP ($R^2 > 0.8$ in the African population group using TOP-LD). Additionally, we looked for variants overlap with potential candidate cis regulatory elements (cCREs) within 100 kb of the lead index variants using visual inspection of regional association plots including LD information. The resources included signatures of promoters, enhancers, and chromatin accessibility (i.e. markers of histone modification, DNase hypersensitivity, CTCF binding, etc.) from BMI-relevant tissues, including blood, brain and liver; from ENCODE.

And, we added the following to the Results section.

“Functional Annotation

To gain a better understanding of the potential functional consequence of the *MTMR3* locus, we used publicly available information from Ensembl VEP, FORGEdb, and ENCODE to annotate nearby variants. Of the 54 variants in high LD with our lead SNP, most were intronic or nearby *MTMR3* (**Supplementary Data 8**). Of these, four variants had a moderate CADD (combined annotation dependent depletion) score (scaled CADD > 10) with rs73394881 having the highest relative CADD score⁶¹, three of which lay within a possible enhancer (rs73396896, $R^2 = 0.884$; rs73394881, $R^2 = 0.889$; rs74832232, $R^2 = 0.889$). There is an abundance of evidence for regulatory action in this region for variants in high LD with our lead SNP. Although our top associated SNP does not directly overlap potential cCREs based on ENCODE, there is limited data of overlapping cCREs from RoadMap data available on FORGEdb (**Supplementary Figure 9, Supplementary Data 8**). Three genes were implicated across these databases, including *MTMR3*, *MTFP1*, and *ASCC2*. Five SNPs were implicated in potential regulation for *MTMR3* and/or *ASCC2* across multiple tissues based on Activity-By-Contact (ABC) data. Additionally, 35 SNPs were significant cis-eQTLs with *MTFP1* in Nerve Tibial cells. Three SNPs exhibited Zoonomia PhyloP scores indicative of accelerated evolution (PhyloP ≤ -2.270), including our tag SNP, rs111490516, and two nearby SNPs, rs57349783 and rs2107673; meanwhile, three that exhibited scores indicative of significant cross-species conservation (PhyloP ≥ 2.270), including rs73396896, rs6006286, and rs73394881. “

As these results implicated two additional genes, we also edited the following text to the Discussion.

“While there is limited knowledge on the biological implications of the lead variant, there is an abundance of evidence for regulatory role for SNPs in high LD with our lead SNP and multiple lines of evidence supporting a role in obesity at this locus. We utilized the Ensembl VEP, FORGEdb, and ENCODE databases to explore predicted functional consequences of our novel locus. Our lead variant is intronic to *MTMR3*, and there is evidence linking SNPs in high LD with regulation of *MTMR3*, *MTFP1*, and *ASCC2*. The primary cellular function of *MTMR3* relates to regulation of autophagy. Although there is no direct evidence linking *MTMR3* to obesity, previous studies have established a connection between *MTMR3* and related cardiometabolic traits. *MTMR3* was associated with LDL cholesterol ($P = 1 \times 10^{-8}$) in a GWAS meta-analysis of European, East Asian, South Asian, and African ancestry individuals. A potential mechanism was proposed later suggesting *MTMR3* may mediate the association between miRNA-4513 and total cholesterol⁶⁸. Furthermore, pyruvate dehydrogenase complex-specific knockout mice with high-fat diet induced obesity also exhibited increased blood glucose and higher expression levels of *MTMR3*⁶⁹. *ASCC2* has no known role related specifically to obesity. However, like previously-implicated BMI-associated genes, it is a ubiquitin-binding protein involved in transcriptional regulation and DNA repair. There is strong evidence for a role of *MTFP1* in regulation of adipogenesis from animal models. For example, liver-specific knockout of *MTFP1* in mice provide evidence for protection against weight gain and ensuing

diseases of metabolic dysregulation (e.g. diabetes and non-alcoholic fatty liver disease). The protection against weight gain appears specific to lipid rich diets through and less pronounced in exposure to carbohydrates. Furthermore, knockdown of *MTFP1* in adipocytes from sheep increased adipogenesis. Decreased expression of *MTFP1* was linked to increased expression of *PPARG* (Peroxisome Proliferator-Activated Receptor Gamma) and *LPL* (Lipoprotein Lipase), two known obesity-related genes important for adipogenesis, lipid metabolism, and ultimately energy homeostasis. Also, differential expression of *MTFP1* was observed during increased adipogenesis and fat deposition in the tail of developing sheep.”

- Regarding previous obesity-related associations nearby our *MTMR3* locus. We did not conduct formal conditional analysis on this SNP as there was no evidence of a shared signal due to low LD ($R^2=0.029895$) and the differences in MAF. We have clarified this in the manuscript in the 3rd paragraph of the Discussion as follows.

“However, this SNP (rs5752989, chr22:29969791) is not in LD of our top associated SNP in this region in our study population ($R^2 = 0.03$), and thus likely an independent signal.”

4. Speliotes et al (2010) reported a secondary signal at *MC4R*. Is the secondary signal reported here independent of the previously reported secondary signal or just a different secondary signal perhaps due to differences in LD? Is it truly new? The authors state that it is not robust to conditional analyses using previously reported variants, which suggests that it does not provide any information beyond the 2 SNPs reported previous in Speliotes or elsewhere. Sup data 9 and 10 should indicated exactly which SNP(s) were used for the conditional analysis. Also, why are some of the SNPs in Sup data 9 different from Table 1? Please clarify.

- Supplementary Data 9 provides the top variants at each locus, after conditioning on the corresponding index variants from our discovery analysis (Table 1). For example, on chromosome 1, rs543874 was the variant that reached genome-wide significance in the discovery cohort. We subsequently tested the associations between BMI and 5,008 nearby variants that were within 500kb of rs543874, while conditioning on rs543874. This internal conditional analysis identified rs111238523 as having the smallest p-value ($P = 4.30E-04$) at this locus; however, this did not meet the significance threshold $P < 5.96E-07$. In Supplementary Data 9, we also provided beta, SE, and P-value for each secondary variants (regardless of statistical significance) before the conditional analysis. To enhance clarity, we have added a column ‘index variant’ and a footnote, indicating the SNPs were used in the internal conditional analysis at each locus.

- In Supplementary Data 10, we focused on the lead SNPs identified in our discovery analysis listed in Table 1 that were not themselves previously reported index variants (marked as ‘No’ in the ‘known index variant’ column). For these variants, we conditioned on known BMI variants from published literature to uncover novel variants independent of established BMI variants. To clarify we have added a ‘known index variant’ column to Supplemental Data 10 to denote the variants used in this known conditional analysis that were within 500 kb of our top SNP. However, I will note that all SNPs in the publications listed in this table were conditioned on regardless of distance.

Also, to clarify within the text, we edited the following under Methods and Results for Conditional Analyses sections.

“To determine whether association signals in known loci were independent of known signals, we performed conditional analyses using previously published index variants. Specifically, for each lead SNP that is not a previously-reported BMI-associated index variant, we conditioned on all known SNPs on each chromosome.”

“We additionally assessed independence for the top variants in known loci, by conditioning on all previously-reported index variants by chromosome, and highlight previously reported index variants located within 500 kb of our top locus SNP (Supplementary Data 10).”

- Speliotes et al. identified rs7227255 as a secondary signal after conducting a conditional analysis on rs571312, the variant most strongly associated with the area near the *MC4R* gene. The SNP rs7227255 was in perfect LD with rs2229616, a rare missense variant. While both rs571312 and rs7227255 are located near but not in *MC4R*, rs2229616 is directly situated on the gene. In our analysis, we conditioned on rs2229616 and 16 other variants on the *MC4R* gene. Our results indicate that our identified secondary signal, rs78769612, is not independent of the other known *MC4R* variants.

To further clarify, we added the following text to the Discussion:

“We also consistently identified multiple association signals of high effect in *MC4R*, which is a well-established monogenic obesity gene, through our discovery analysis, internal conditional analysis, and rare variant aggregate analysis. Despite not identifying novel SNPs in *MC4R* that are independent of known BMI-associated SNPs, we replicated the secondary signal in this gene region, rs2229616, a rare missense variant previously reported in individuals of European ancestry by Speliotes et al.”

5. There have been a number of SNPs and genes from exome sequencing reported for BMI previously (e.g., Akbari, Science 2021). Were these SNP and gene-based associations from Akbari et al replicated in this study? The authors did aggregate testing and said that they replicated the gene-based *MC4R* finding, but what about the findings from Akbari? Given the number of different populations in the dataset, evaluating these previously report SNPs across populations seems like a missed opportunity.

- Thank you for pointing this out and we concur that omitting a comparison would be a missed opportunity. While we replicated *MC4R* but not the other 15 genes from Akbari et al., we have included the results of our rare variant aggregate tests for all 16 genes from Akbari et al in the updated Supplementary Data 11. Accordingly, we have expanded the Aggregate-based Testing section in the Results as follows:

“We did not identify any novel gene regions through association tests at the genome-wide level ($P < 5 \times 10^{-7}$) when aggregating variants with $MAF \leq 1\%$. Nevertheless, we successfully replicated previous gene-based associations with the well-known melanocortin 4 receptor (*MC4R*) gene ($P = 8.47 \times 10^{-8}$), with 110 alleles across 37 sites within coding regions, enhancers, and promoters for *MC4R*. The *MC4R* locus was also identified in single-variant analyses. We also examined the 16 genes reported by Akbari et al. with an exome-wide significant association with BMI (Supplementary Data 11). Although only *MC4R* reached genome-wide significance in our aggregate-based testing, three additional genes – *ROBO1*, *GPR151*, and *ANO4* – showed nominal significance ($P < 0.05$) in our SMMAT test.”

We further revised the Discussion on page 25 as follows:

“In addition to *MC4R*, three other genes – *ROBO1*, *GPR151*, and *ANO4* – of the exome-wide significant genes identified in Akbari et al. showed nominal significance in our rare variant aggregate analysis. This lends further support to the generalizability of these genes across populations, given that 85% were of European ancestry and 15% of admixed American ancestry in Akbari et al., compared to our more diverse cohort with 49% European and 51% other populations.”

6. Fine mapping was done using the multi-ancestry summary statistics with PAINTOR, but such an analysis does not fully capture the LD structure of the different populations included and highly dependent on the functional annotation. The functional annotation based on GeneHancer does not appear to be tissue or cell-type specific, which may limit its ability to accurately fine-map the loci. Were other annotations considered and tested, such as regulatory elements from neuronal tissue?

- As our primary analysis was a multi-population analysis, we used the LD calculated from our analytical sample to perform all fine-mapping analyses. This was done to preserve the LD structure that was used to identify the association signal. Additionally, previous analyses, including GWAS of BMI (see Lockett et al 2015), have shown that fine-mapping in multi-population analyses can improve results (i.e. decrease the window of credible sets) by leveraging differences in LD patterns across populations.

- It is true that the Genehancer data used was not tissue-specific, which may limit some potential inferences that are tissue or outcome specific. However, the Genehancer annotation utilized herein was generated as part of a larger TOPMed effort to provide a harmonized database with potential broad applications that could be used across the TOPMed Program projects and included all variants in the TOPMed study populations for the data freeze. Thus, we felt that leveraging the harmonized data resource allowing for cross-study comparisons is a benefit that outweighed some potential limitations.

7. Not much detail is given regarding the assessment of BMI. Were weight and height measured in the participants or was the data obtained through self-report?

Each cohort has previously published manuscripts and/or publicly available protocols on measurement standards through study websites and/or dbGAP. While >90% of our data comes from studies with measured height and weight, there are a few studies that relied on self-reported weight (e.g. WGHS), a mix of measured and self-report (e.g. WHI), and measures derived from electronic health records whereby weight may be measured and height self-reported (e.g. BioMe). We acknowledge there is a possibility of heterogeneity being introduced by including participants with self-reported weight and/or height, which could diminish power for discovery by increasing phenotypic noise.

8. Minor clarification on methods (page 13): The authors state that LDlink was used to calculate pairwise LD between potentially independent signals in known loci and produce LD heatmaps using 1K data. In other parts of the paper, TopMed was used to estimate LD. Why wasn't the data from TopMed used to estimate pairwise LD and produce heatmaps?

LDlink was used for two reasons. We wanted to examine LD in known association regions based on reference panels that would have been used to impute previous GWAS studies, and for visualization of the LD patterns for the region. The LDmatrix Tool on LDLink does not currently provide the option of uploading your own LD data to allow for side-by-side comparisons. To make this clearer, we have edited the methods as follows.

“For visual inspection and calculation of LD in reference panels used for imputation in previous GWAS studies, LDlink was used to calculate pairwise LD between potentially independent signals in known loci and produce LD heatmaps using the 1000 Genomes Global reference panel.”

Reviewer #2 (Remarks to the Author):

This manuscript describes a cross-ancestry genetic analysis of BMI focusing on non-European cohorts. For the most part it identifies known BMI associations, and it also identifies a potentially novel association with the variant rs111490516 in *MTMR3*, which is fixed in individuals with European genetic similarity and has various frequencies in other populations.

It is nice to be able to show that analyzing beyond European genetic similarity can identify novel results. I am still a little torn about whether or not the *MTMR3* hit is real, but overall the authors have done a good job of investigating from as many avenues as possible to try to show that this result holds up.

I would advise the authors to use the phrasing "genetic similarity" in their paper following the latest recommendations <https://nap.nationalacademies.org/catalog/26902/using-population-descriptors-in-genetics-and-genomics-research-a-new>.

- We thank the reviewer for advising language that is consistent with the NASEM. In this paper, we followed the TOPMed recommendations on the use and reporting of race, ethnicity, and ancestry in genetic research (PMID: 36119389), and used “genetic ancestry” to describe genetic origins and “race” and “ethnicity” to refer to non-biological social categories (i.e. “European ancestry” for referencing genetic ancestry and “White” for race). Additionally, it is important to note that the majority of our participants were self-identified and only 10% were HARE-imputed which is a combination of genetic similarity and self-reported population group. As a result, our particular designation of population groups does fit within the guidelines for use of genetic similarity, thus we have chosen to retain the use of “population groups” throughout. We hope that our description of how these population groups were defined upholds the values of transparency and fits within the purview of the “group label” designation outlined in the NASEM (“name given to a population that describes or classifies it according to the dimension along which it was identified”), while still respecting the guidelines of the NHLBI TOPMed Program. We have made sure that these terms are used consistently throughout the manuscript to help alleviate any misinterpretation, and all instances of clarification are highlighted in red.

While it is becoming more common to analyze data across populations, especially in collapsing analyses of rare variants, I'm not sure the field is convinced that these types of analyses are always appropriate when analyzing individual variants. It is still typical to see analyses performed on each ancestry separately, and I was glad to see the European and African separate analyses here to shore up the

results. I would however like to see more citations and explanations of why it's ok to analyze all the ancestries together and what reassurances there are that biases are not occurring.

- This is a very important question, which was also raised by reviewer #1. Please refer to our response to the first comment from Reviewer #1 comments 1 and 2 for an explanation of our rationale behind selecting a pooled analysis approach. Additionally, as requested, we have expanded our references in the Discussion to include those mentioned in our response, specifically, PMIDs 26748518, 31217584, 36119389.

Also, in the European and African-specific analyses, I was not sure if new PCs specific to those genetic similarity groups had been generated and used for the analysis, or if the PCs that were generated for the whole group were what was used still. In a group-specific analysis, PCs generated specifically for that genetic similarity group should be used.

- We did not generate European- and African-specific PCs and instead used the PCs generated from the total study population in both population-specific analyses. By projecting at a larger population level, the generated PCs would still capture the diversity while allowing for cryptic relatedness across population groups and studies.

However, when discussing the group-specific analyses, the authors indicate that rs111490516 was not the genome-wide significant hit in *MTMR3* in the African analysis, and a different variant rs73396827 is associated instead. This needs more exploration. What was the p-value for rs111490516 in this analysis? What was the p-value or beta for rs111490516 and rs73396827 in the other genetic similarity groups included in the cross-ancestry analysis? There is a history in human genetics of putting too much emphasis on replication that is not actually replication, i.e., seeing a signal with another variant in the gene instead of the original one and not properly correcting for how many variants in the gene were tested. Here it sounds like the variants are in very high LD in the African genetic similarity group, so this shouldn't be an issue, but it is helpful to explain further what is going on.

As the reviewer indicates, the top variant in our population-group stratified analysis differs from the multi-population analysis, but is in strong LD. In the African population group analysis, rs111490516 exhibits a *P*-value very close to, but not as significant as, that of rs73396827 (see below). While we did not conduct stratified analyses in all population groups for the reasons outlined in our response to reviewer 1, we provide additional details for clarification and in support of our findings.

First, to clarify the difference in association results for the two lead SNPs at our novel locus, we provide the results below.

In the pooled analysis:

- rs73396827: beta = 0.0781, SE = 0.0133, *P* = 4.86e-09
- rs111490516: beta = 0.0783, SE = 0.0133, *P* = 4.52e-09

In the African-specific analysis:

- rs73396827: beta = 0.0873, SE = 0.0145, *P* = 1.66e-09
- rs111490516: beta = 0.0870, SE = 0.0145, *P* = 1.82e-09

Additionally, to improve confidence in the association results for this locus, we conducted a replication of rs73396827 using the same replication cohorts (for MVP, we used their latest release), plus the inclusion of the MyCode Community Initiative study. Additionally, we provided details on the LD between the two variants in each of our replication cohorts to show that ubiquitously, these two SNPs are in high LD and likely represent the same association signal.

The results have been added to **Supplementary Data 7, Figures 3**, and we have added **Supplemental Figure 8**, a forest plot of rs73396827.

Accordingly, we have updated the Results section on Replication as follows:

“The replication sample sizes ranged from 3,213 in MyCode to 80,730 in MVP (**Supplementary Data 7**). In the six replication studies of Blacks, Africans, and African Americans, rs111490516 and rs73396827 in *MTMR3* displayed high LD ($R^2 = 1.00$) (**Supplementary Data 7**). Their MAFs are nearly identical across studies and ranged from 11% to 13%, aligning with the 13% observed in our African and Barbadian groups (**Supplementary Data 7**). We replicated both variants, demonstrating directionally consistent associations with BMI across the replication studies. In MVP and REGARDS, particularly, both variants reached statistical significance at $P < 0.05$. We observed a 68% reduction in the estimated effects when meta-analyzing across replication studies (rs111490516: $\beta = 0.025$, $SE = 0.007$, $P = 1.92 \times 10^{-4}$, $MAF = 12\%$; rs73396827: $\beta = 0.026$, $SE = 0.007$, $P = 1.50 \times 10^{-4}$, $MAF = 12\%$), compared to the discovery analysis (**Figure 3, Supplementary Data 7, Supplementary Figure 8**). In the meta-analysis of up to 202,675 individuals from both discovery and replication studies, rs111490516 and rs73396827 both had an estimated effect of 0.36 with a SE of 0.006 and P -values of 2.40×10^{-9} and 1.60×10^{-9} , respectively (**Figure 3, Supplementary Data 7**).”

We further revised the second paragraph in the Discussion as follows:

“Similarly, in our replication analysis of 113,802 individuals with imputed genotypes, rs111490516 was suggestively significant ($\beta = 0.025$, $P = 1.92 \times 10^{-4}$, $MAF = 12\%$). In the African-specific analysis, although rs111490516 was genome-wide significant ($\beta = 0.087$, $SE = 0.015$, $P = 1.82 \times 10^{-9}$), rs73396827 was the top hit at this locus. These two SNPs further exhibited absolute LD, as well as consistent directions of effect, SEs, and P -values across replication studies and in the meta-analyses. Therefore, the lack of discovery in prior publications may not be due to insufficient power.”

Furthermore for the *MTMR3* variant, in table s7, it is shown that the beta is lower and the pval nowhere near significant in the individual replication groups, before the meta analysis. I know there is winners curse, but the similar sample size of MVP at least to the discovery cohort and the substantial difference in beta and pval of 0.04 despite the allele frequency being 3x higher in MVP, and therefore should have better power, is rather concerning. Do the others have a hypothesis for why the signal is so questionable in MVP in particular? Is it because the imputation likely did not go well? Overall, if imputation accuracy is a concern, can the authors use the UKB WGS data instead of imputation to help recover the signal? For me, Table S7 and especially the MVP numbers were the part of the paper that made me the most hesitant about the finding. It may still be a real association, but we need more explanation for what is going on here. The explanation given in the second paragraph of the discussion that the lack of clear replication may be because it's not clear which variant is causal in the region is not very convincing to me.

- We hope that by performing replication of rs73396827 in studies of similar population background would ease the reviewer's concern and our updated text help clarify the situation. As our replication studies restricted to samples of self-identified Black, African American, or African genetic similarity in their study, the allele frequencies for our variants were similar to that observed in our African population group analysis, but as noted by the reviewer, the betas were reduced likely due to winner's curse.

Additionally, it is true that the MVP sample size is closer to the discovery sample but with an even greater reduction in beta value compared to other studies, which we attribute to decreased imputation quality. As illustrated in Supplemental Data 7, all other studies imputed to different imputation reference panels, all had higher imputation R^2 (0.88 in MVP and 0.98-1.00 in other cohorts, *including UKBB*) and all exhibited higher beta values compared to MVP (0.0183 and 0.0185 in MVP for rs111490516 and rs73396827, respectively compared to 0.0254-0.0585 for other studies), with the highest betas observed in those studies imputed to the TOPMed reference panel.

In our updated replication analysis, we utilized the latest release of the MVP data (Release 4), contrasting with the previous release used in the submitted manuscript. In this latest data set, imputation was performed to a hybrid reference panel comprised of the African Genome Resources panel (Sanger Imputation Service, available at <https://imputation.sanger.ac.uk/?about=1#referencepanels>) and 1000 Genomes Project (p3v5), as detailed in the MVP Release 4 paper (PMID: 37425708). With an improved imputation panel, the imputation score for both rs111490516 and rs73396827 in the MVP African American population reached 0.88. While there was not a great increase in sample size ($N = 79,889$ to $N = 80,730$), there was a modest improvement in imputation accuracy in this sample ($R^2 = 0.71$ to $R^2 = 0.88$) and an increase in the association (Beta = 0.0181 and $P = 0.0442$ to Beta = 0.0183 and $P = 0.0262$). MVP has previously pointed out that there are differences in imputation accuracy across population groups within MVP due to array design and reference panel, and are still working on an "optimized imputation strategy. The strategy implemented by the UK Biobank was to use the Haplotype Reference Consortium (HRC) as the main imputation reference panel and to supplement with variants from 1000 Genomes if those variants were missing in HRC. However, this is not viable for MVP, which has a large proportion of non-European individuals. Further understanding and analyses of imputation strategies in a multi-ethnic and admixed cohort will be required if we are to obtain an optimized strategy for MVP." (PMID: 32243820).

A small thing, one phrase bothered me: "In the two population-specific analyses, 10 association signals reached genome-wide significance (Supplementary Data 6, Supplementary Figs 4 – 7). Two of these signals were also detected in the multi-population analysis." What are the other 8 hits, are they known hits? Why don't they come up in the cross ancestry analysis? This appears better explained in supplementary data 6 that they are in LD with hits in the main analysis, but in that case this sentence should be rephrased I think.

- We thank the reviewer for pointing out this typo. We have edited the sentence for clarity as follows:

"In the two population-specific analyses, 10 association signals reached genome-wide significance (Supplementary Data 6, Supplementary Figs 4 – 7). All 10 association signals were identified in the multi-population analysis, including two of these loci with the same index SNPs in at least one population, *SEC16B* in African and *FTO* in European. For these same two loci, each population-specific

analysis also revealed a distinct lead variant compared to the multi-population analysis; ...The remaining SNPs were in proximity to and in LD with the index SNPs in the corresponding loci from the multi-population analysis.”

Reviewer #3 (Remarks to the Author):

Whole genome sequencing analysis of body mass index identifies novel African ancestry-specific risk allele

This manuscript by Zhang, Brody, Graff, Highland, Chami, et al. reports on a study of whole genome sequencing data from among almost 90,000 participants of high ancestral diversity when compared to the vast majority of genetic datasets in the field. The authors analysed these data to systematically investigate genetic associations with body mass index (BMI). Of the index associations they find, one was novel, and common among individuals of African descent. After finemapping, they report two putatively causal variants in *POC5* and *DMD* loci.

The study was generally carefully conducted, and made use of a variety of approaches to interrogate the genetic data. Importantly, the authors went to the trouble of replicating the association found in the discovery sample through a meta-analysis of MEC, MVP, BioMe, UKBB, and REGARDS. I also applaud the authors for reporting the negative result in the rare-variant association testing. The technical execution of the work appears robust, and the study is well-written. However, I do have a few reservations:

- We thank the reviewer for their appreciation of our robust approach to discovery, interrogation, and reporting of findings. We have responded to each of their remaining concerns below.

Major comments

1) A potential issue of the study is that the title implicitly suggests that the novel association would not have been found without whole genome sequencing data. The authors allude to this in the discussion: “our index SNP [rs111490516] is likely not causal but could be in LD with a causal SNP and also poorly captured in studies relying on imputation”. On its face this is very interesting, and warrants investigation: is the variant poorly imputed because of European focused SNP array design not allowing the variant to be imputed? Is it because we are lacking imputation panels that would allow for better imputation of array data from individuals of African descent? That said, Supplementary Data 7 seems to suggest that current imputation panels do quite a good job of imputing the variant, so do we really require sequencing data to determine the genotype at rs111490516 accurately? It seems that this question could be answered by taking the discovery set (TOPMED), masking the variants not present on a set of SNP arrays (e.g. that used for MVP, given that it displayed the lowest R²/info score), and then determine whether the association signal would have been recapitulated using HRC or 1000G as a reference panel. If the signal could not be recapitulated using this approach, it would lend a huge amount of evidence to one of the central assertions of the paper.

- This is indeed a difficult situation to clearly discern. Adding to the complexity is the various arrays used across previous GWAS studies and combinations of imputation panels previously used. Unfortunately, due to the data restrictions (TOPMed data rested on Analysis Commons within DNAnexus during time of analysis and now exist on BioDataCatalyst), restricting to a subset of variants and imputing back to various reference panels that are not housed within the same databases is not within the scope of this study.

However, to assist in addressing this question, we did examine the imputation quality for our sentinel SNP for studies in TOPMed and PAGE that we have locally and have been imputed to both TOPMed R2 and 1KGP3 global reference panels. In total, we have data on 21 different data sets separated by self-reported race/ethnicity, study, and genotyping array and combined for samples typed on the same array. Across all data sets the mean imputation pseudo R^2 was higher for TOPMed R^2 (0.972) compared to 1KGP3 (0.876); however, we did observe differences in imputation quality across different strata. For example, for studies typed on the Illumina MEGA array, imputation quality was always better for 1KGP3 reference panel, except in the case of American Indians. When looking by race/ethnicity, imputation quality was better for TOPMed R2 reference panel for Black/African Americans and Hispanic/Latinos for more studies (3:2), an equal number of studies across the White/Europeans (4:4) and when imputation was performed in a pooled sample (1:1).

Given that we are unable to completely tease this apart at this time, we have edited discussion to decrease the strength of our assertion.

“Therefore, the lack of discovery in prior publications **may not be** due to insufficient power. As indicated by our fine-mapping results **and potential regulatory role of nearby variants** in this novel locus, our index SNP is likely not causal but could be in LD with a causal SNP and also poorly captured in studies relying on imputation. In other words, the causal variant underlying this locus may be nearby, less frequent, and on an LD block more frequent in a population poorly represented in other imputation reference panels, but well represented in our WGS and highly diverse sample (e.g., Caribbean admixed individuals). In this case, one would require sequencing data in a large sample size with the relevant haplotype to detect a significant association that was not able to be identified with imputation in a similar number of people. **Thus, future studies with WGS in study populations with genetic similarity and functional follow-work are needed to further narrow in on the causal variant(s) underlying this association signal.**”

Additionally, to support our claim that our lead SNP is likely not causal, but in LD with a more likely causal SNP, we have added additional functional annotation and illustrations of purported regulatory roles for SNPs in high LD with our lead variant (see **Response to Reviewer 1, number 3, Supplementary Data 8 and Supplementary Figure 9**).

2) As it currently stands, the quality control and analyses in the paper are not reproducible. While I understand that detailed descriptions of the methods cannot be provided in detail in the main text due to length restrictions, they should be present in complete detail in the supplement. For example, what thresholds have been used in HARE? How were PCs defined? How many PCs were used in the analyses? How was sex concordance defined? In pheWAS, what thresholds were used to define ‘African’ ancestry? Etc, etc.

Also, there is no code provided to reproduce the analyses. Can you please link to the code used for the QC and analyses in a publicly available repository (e.g. on github)? Upon clicking the link to find the

methods for quality control on the TOPMed website, I get 'access denied' and told to email regarding membership. These methods could be extremely valuable to the genetics community, and would have the greatest impact and use if they were not only available to members of TOPMed. Sharing of code and detailed methods would be particularly impactful given the need for careful analysis of data from diverse ancestries.

- We thank the reviewer for highlighting some specific areas where additional analytical details would be helpful for clarification and reproducibility. In addition to addressing these specific areas, we have made clarifications and corrections throughout the methods section. Also, we have added a "Data and Code Availability" section to our manuscript. All edits to methods are highlighted in red within the text, and we highlight a few of the specific changes requested by the reviewer below.

For the HARE analysis, we used the first 9 PC-AiR PCs generated on this subset of samples used in the analysis. These 9 PCs were selected based on the elbow method and a scree plot. Participants with self-reported population group were used to train the HARE model, while those with unreported or non-specific self-reported group were excluded from the training step. We did not apply a specific threshold to the posterior probability of membership in the final step, but assigned those with unreported or non-specific self-reported group to the existing population groups with the highest HARE probability. All other participants remained in the population group assigned based on their self-reported race/ethnicity/population group. We have clarified this in the methods as follows:

"For individuals who had unreported or non-specific population memberships (e.g., "Multiple" or "Other"), we applied the Harmonized Ancestry and Race/Ethnicity (HARE) method to infer their group memberships using genetic data, **excluding these individuals from the training step**. This imputation was applied to 8,015 participants (9% of the overall population), assigning each to one of the existing population groups **based on the group with the highest probability of membership**. **All other participants remained in the population group assigned based on their self-reported race/ethnicity/population group**. In this way, our study population groups were defined based on a combination of self-reported identity and the first nine genetic principal components (PCs) (**Figure 1, Supplementary Fig 1, and Supplementary Data 1**). **The decision to use nine PCs was informed by the elbow method and scree plots.**"

In regard to sex concordance, genetic sex was determined using the ratio of X and Y chromosome to autosomal depth. A small number of sex mismatches were detected based on comparisons to study-reported sex. These samples were either excluded from the sample set if not resolved. Additional details can be found here: <https://topmed.nhlbi.nih.gov/topmed-whole-genome-sequencing-methods-freeze-8#concordance-between-annotated-sex-and-genetic-sex-inferred-from-the-wgs-data>.

We apologize for the difficulty in accessing the TOPMed WGS QC protocols. They are currently publicly available on the TOPMed website (<https://topmed.nhlbi.nih.gov/topmed-whole-genome-sequencing-methods-freeze-8>). For each step, a link to a github repository with relevant code is provided. We have included this exact URL in our **Data and Code Availability** in addition to other direct links.

"Data and Code Availability

The summary statistics from our GWAS analyses presented are provided through the NHGRI-EBI Catalog of human genome-wide association studies (<https://www.ebi.ac.uk/gwas/>). TOPMed Program individual-level data is available through Google and AWS cloud services following NIH dbGap approval. Details on gaining access are found on the TOPMed website (see

<https://topmed.nhlbi.nih.gov/topmed-data-access-scientific-community> and <https://topmed.nhlbi.nih.gov/topmed-whole-genome-sequencing-methods-freeze-8#access-to-sequence-data>).

All protocols used for variant calling and quality control for data used in this study are described here: <https://topmed.nhlbi.nih.gov/topmed-whole-genome-sequencing-methods-freeze-8>. Links to relevant code, including github repositories, are also provided on the same site.

All GWAS analyses were performed using GENESIS, a bioconductor package, on the TOPMed Analysis Commons. Details of the GENESIS app used on the Analysis Commons, along with underlying package information and code are provided here: https://github.com/AnalysisCommons/genesis_wdl."

Last, we have added the following to the PheWAS section in the Methods: "PheWAS were restricted to African Americans in each study: **In MyCode, African Americans were identified through** electronic health records (a combination of self-report and clinician-reported race/ethnicity), while in BioME the identification was based exclusively on self-report."

3) Availability of results. It would be great to see full variant and gene-based association test statistics made available to the community. As it stands, there is currently no data availability statement for the analysis results. Will all summary statistics be made publicly available?

TOPMed and our group are very supportive of the public availability of data, including summary results. As noted in the above Data and Code Availability section we have added, results will be shared on the GWAS Catalog upon assignment of a PMID following acceptance of the manuscript.

4) The claim that "no previous study has successfully identified BMI risk variants of high confidence at the POC5 and DMD loci" in the discussion appears to be immediately contradicted "these two variants were also considered high-confidence causal variants (PPrs2307111 = 0.96, PPrs1379871 = 0.99) in a recent joint analysis of three biobanks (UKBB, FG, BBJ)". Is this statement specifically referencing the analysis of non-European populations? Further, the claim that because the variant is at high prevalence across multiple populations implies that it is more likely to be causal seems speculative and should be backed up. If this is what you are claiming, why should we conclude this?

- We thank the reviewer for highlighting the apparent contradiction. The referenced paper (Kanai et al.) is currently a preprint and has not yet undergone peer-review. To clarify, we have revised our statement as follows:

"While there have been multiple attempts to fine-map previously identified BMI loci, no previous **peer-reviewed and published** study has successfully identified BMI risk variants of high confidence at the *POC5* and *DMD* loci. By applying a Bayesian fine-mapping approach, we reduced associated signals to 95% credible sets of two likely causal SNPs. Functional annotation revealed that one of them, rs2307111, was a benign missense variant in *POC5* (NP_001092741.1:p.His36Arg) according to ClinVar, while the other is an intron variant in the promotor region of *DMD*. **Nevertheless**, these two variants were also considered high-confidence causal variants (PP_{rs2307111} = 0.96 in UKBB, PP_{rs1379871} = 0.99 in both UKBB and FG) in a recent **preprint of a** joint analysis of three biobanks (UKBB, FG, BBJ).

- Additionally, we understand the concerns raised regarding the interpretation of the variants' frequencies across multiple populations. It is important to clarify that our intention was not to imply a causal relationship based solely on this observation. Rather, our aim was to underline a notable pattern in the prevalence of the variant, which we believe warrants further investigation. With this in mind, we have revised our discussion to make this distinction clear and avoid any speculative implications:

“Notably, unlike in Kanai et al. where the PP appeared to be driven by the Europeans (for rs2307111: $PP_{UKBB} = 0.96$, $PP_{BBJ} = 0.12$, $PP_{FG} = 0.01$; for rs1379871: $PP_{UKBB} = 1.00$, $PP_{FG} = 1.00$), the effect alleles in our study were observed in high proportions across many non-European population groups (**Supplementary Data 5**).”

Minor comments

1) Why do you use 9 PCs in HARE? Looking at the paper, they suggest 30. What metric did you use to determine that 9 was the appropriate cutoff? Scree plot?

- The decision to use 9 PCs in HARE was informed by examining scree plots produced within our analytical sample. We clarify this in the manuscript as noted above (concern 2).

2) References are missing for the various annotations used in the rare variant association analyses (LOFTEE, MetaSVM, FATHNMM-XF). Can the authors outline why these particular metrics were used in preference to others, particularly given the recent influx of damaging missense prediction metrics?

- We have added references for annotation tools. The choice at the time of analysis was largely informed by this paper: <https://www.ncbi.nlm.nih.gov/pmc/articles/PMC4752381/> and based on high agreement and performance. We have added this reference to our manuscript for clarification and edited the text as noted below in the “Rare Variant Aggregate Association Analysis” section of the Methods. LOFTEE was selected by the data coordinating center (DCC) for TOPMed as part of the centralized and harmonized annotations and we used their work for those annotations.

“We chose to focus on annotation that was previously shown to have high agreement with other annotation resources and high prediction accuracy.”

3) For the rare variant collapsing analysis, it would be interesting to see more of a deep-dive into the different classes of variant and a combined analysis e.g. via a Cauchy combination test, as in other publications using e.g. SAIGE-gene or REGENIE. As it stands, although the results are null, it seems like a missed opportunity for a deep investigation of rare-variant group based testing. Did any other (non-novel) genes pop up in the gene-based analysis, or was it only MC4R?

- Although only *MC4R* reached statistical significance in our gene-based analysis, to provide further context, we have compared our gene-based results to that in Akabari et al. (PMID: 34210852) by (1) introducing a supplementary table summarizing our SMMAT results for the 16 genes they reported (see

new Supplementary Data 11), and (2) incorporating relevant discussion in the Results and Discussion sections; please refer to our response to Reviewer #1's comment 5 for further clarification.

We appreciate the reviewer's suggestion on further investigating rare-variant based testing. As TOPMed has been continuously expanding, we do indeed plan to repeat these analyses and expand our protocols in the TOPMed anthropometric working group and specifically target rare variant analyses as power for discovery improves. However, we note that GENESIS and not SAIGE not REGENIE was chosen to its analytical flexibility in relationship to allowing for heterogeneity of effect by population group, an option well-suited to the demographic and genetic background of our study population. Also, as new resources, including annotation and analytical tools, are constantly evolving, we will certainly continue to rethink our protocols moving forward.

We are very excited to release this first wave of WGS analyses of BMI and are looking forward to potential secondary analyses that can be achieved with our summary results, including post-hoc gene-based analyses from groups with alternate perspectives.

4) In pheWAS, only removing up to second degree relatives seems permissive (assuming that a mixed model was not used to run the association at the locus). Are the authors sure that the analysis would be robust if they had detected a pheWAS hit? Given the restriction in bioME to samples with at least 20 cases or controls, that would imply that a large number of the phenotypes will have large case-control imbalance. It would be helpful to know what tests were used, and that the association test statistics were checked for robustness.

- Between the two studies, the MyCode study has a higher degree of relatedness. While not performed for this specific analysis, our group has previously performed PheWAS analysis restricting to 1st degree, 2nd degree, and 3rd degree unrelated individuals, and have produced largely concordant results. Additionally, as our results were restricted to self-reported Black/African Americans there are few relatives within our dataset beyond 2nd degree. We restricted to unrelated individuals and PheCodes with a minimum of 20 cases and 20 controls in both studies as we implemented standard logistic regression in the PheWAS R Package. As we did not identify any phenome-wide significant results, we did not conduct any sensitivity analyses. Additionally, our top associated phecodes (sleep apnea and obstructive sleep apnea) are common traits with a prevalence of ~20% in our studies and thus controlling for case-control imbalance is not recommended here.

5) The analysis would benefit from a more thorough investigation into the functional consequences of the finemapped SNPs. As currently written, it seems like a bit of an afterthought.

- We thank the reviewer for this suggestion. We have expanded Supplementary Data 8 to include relevant functional annotation from VEP and ForgeDB as was done for our novel locus. We have edited the methods for fine-mapping to add the following sentence, **"To gain a better understanding of the potential functional consequence of likely causal variants within each locus, we used Ensembl Variant Effect Predictor (VEP)²⁹ and FORGEDb³⁰ to annotate all variants with posterior probabilities (PP) >0.5."**

We have also modified the results for fine-mapping as follows:

“Of the seven variants with PP above 0.50, all have evidence of moderate to high impact on a nearby gene. All had FORGEdb scores of 5 or above and six had CADD scores greater than 15 for at least one transcript (**Supplementary Data 8**). For example, the causal variant implicated in our fine-mapping for the *FTO* locus, rs1421085, is intronic or within an enhancer for *FTO*, has a FORGEdb score of 9 and scaled CADD score of 19.58. Multiple studies on this known obesity-risk variant have confirmed this as a causal variant in the pathogenesis of obesity. Also, rs55731973, in the *ZC3H4* locus, while not our index SNP, has a FORGEdb score of 10 (the highest score) and scaled CADD score of 18.84 due to multiple lines of evidence that it sits in an active regulatory site across multiple tissues. None of the seven exhibited a CATO score >0.1, indicative of strong cross-tissue evidence of transcription regulation, a conservative metric for identifying high confidence regulatory variants. Four of the seven SNPs exhibited high Zoonomia PhyloP scores indicative of significant cross-species conservation (PhyloP \geq 2.270), including rs1379871, rs2307111, rs55731973, and rs1421085.”

And, the following in the Discussion centered on fine-mapping results:

“Beyond these two high confident SNPs, five additional SNPs in as many loci were identified in our fine-mapping analysis with a PP>0.5, all with moderate to high evidence of a regulatory or functional role related to a nearby gene. Given that one of these SNPs, rs1421085 in *FTO*, has been successfully confirmed as a causal variant at this locus, additional variants highlighted in our fine-mapping analysis warrant consideration in future functional studies.”

6) It would be helpful to know from the MVP investigators why rs111490516 was removed in their quality control pipeline.

- The MVP investigators used data from an earlier release, while the results provided here is from the latest release that included rs111490516. We have clarified in our Discussion on page 23 as follows:

“It is not available for lookup in the most recent MVP BMI GWAS²³, although included in our replication with the latest MVP data release (PMID: 37425708).”

7) How do you arrive at 18 variants? Are the remaining two variants over the 16 reported in the single-variant analysis section those that are in tight LD with the index SNPs in the multi-population analysis? If so, should they really count as an additional two associations?

- We apologize for any confusion. We arrived at 18 variants by combining the 16 top SNPs and the 2 conditionally independent secondary signals from the internal conditional analysis, as shown in Table 1 and Supplementary Data 9. We have clarified in our text in the Abstract as follows:

“We discovered 18 BMI-associated signals ($P < 5 \times 10^{-9}$), including two secondary signals.”

8) PAINTOR now allows multiple causal associations at a locus. Why was a single causal variant assumed in finemapping? If the assumption was relaxed, could that be used to better ascertain the likely causal variant in the locus, and also to check robustness of the implicated variants when assuming a single

causal variant? It may also be beneficial to compare and contrast the results with other methods e.g. SuSiEx.

- We assumed one causal variant at each locus, as PAINTOR, and other fine-mapping methods that allow this option become less reliable when greater than one causal variant are assumed, but only one exists. Compared to other programs, PAINTOR performs as well or better (especially with annotation) when one causal SNP is present and assumed, but performs worse the greater number of causal variants are present (see: PMID: 25948564 and 34543273). SuSiEx is beneficial when stratified results are used, which is not the case here and the method was still not peer reviewed at the time of our paper (medRxiv 2023.01.07.23284293).

In the preprint led by Kanai et al (mentioned earlier), they performed fine-mapping using both FineMap and SusiE and then combined results from these two methods by taking an average of posterior inclusion probabilities (PIP), excluding variants with a substantial PIP difference (> 5%) to further improve fine-mapping accuracy. By using such a method that is different from ours, they also reported the same SNPs in the *POC5* ($PIP_{\text{susie}} = 0.96$, $PIP_{\text{finemap}} = 0.97$; both in UKBB) and *DMD* (in FinnGen: $PIP_{\text{susie}} = 1.00$ and $PIP_{\text{finemap}} = 1.00$; in UKBB: $PIP_{\text{susie}} = 1.00$ and $PIP_{\text{finemap}} = 1.00$). Thus, PAINTOR performed similarly as these other two methods.

For our two loci that do have evidence of multiple signals following conditional analyses, the secondary signals are >100kb from our index SNPs, and thus these results were not included. As noted in the SuSiEx preprint (medRxiv 2023.01.07.23284293), PAINTOR (and other fine-mapping software) is not scalable to large sets of summary statistics, and thus, with WGS data, we are limited on the size of the window for which we can fine-map. As new methods are rapidly developing, it will become more computationally feasible to consider a full window of association using WGS.

Last, given the suggestion of the reviewer, we did perform and examine fine-mapping using PAINTOR on our pre-defined windows used for a single causal SNP (Response Table 2). As expected, results were largely inconsistent between the two runs, likely due to sensitivity to misspecification of the number of causal variants, as noted before by others. Here we report each index variant, which was the highest probability causal variant in PAINTOR analysis assuming one causal variant, in 14 of 16 loci tested. Assuming two causal variants, the index SNP (italicized below) is among the most probable in 4 of these. Of the two loci with a SNP exhibiting a $PP > 0.9$, one of them (*DMD*, rs1379871, X:31836665) maintained a $PP > 0.9$ assuming 2 causal signals.

Response Table 2. Summary of credible sets following fine-mapping with PAINTOR and assuming 2 causal signals.

Locus rsID	CHR	POS	REF	ALT	EAF	MAF	PAINTOR Annotation	Beta	SE	P-value	PVE	PP
rs543874	1	177935293	T	C	0.421	0.421		0.035	0.005	4.85E-12	0.0005	1.000
rs543874	1	177920345	A	G	0.204	0.204		0.064	0.006	1.38E-26	0.0013	0.000
rs543874	1	177935616	T	C	0.537	0.463		-0.032	0.005	6.13E-11	0.0005	0.586
rs939584	2	638918	A	C	0.852	0.148		0.056	0.007	1.62E-16	0.0008	1.000
rs939584	2	621558	C	T	0.852	0.148		0.058	0.007	1.99E-17	0.0008	0.000
rs939584	2	650980	G	A	0.149	0.149		-0.055	0.007	7.17E-16	0.0007	1.000
rs10182181	2	24940914	T	C	0.577	0.423		0.029	0.005	2.79E-08	0.0003	1.000
rs10182181	2	24927427	A	G	0.561	0.439		0.035	0.005	1.76E-11	0.0005	0.000
rs10182181	2	24964730	C	T	0.424	0.424		-0.029	0.005	3.28E-08	0.0003	1.000
rs869400	3	186114625	C	T	0.172	0.172		-0.037	0.006	1.04E-08	0.0004	1.000
rs869400	3	186108951	T	G	0.820	0.180	genehancer	0.038	0.006	1.21E-09	0.0004	0.000
rs869400	3	186116710	A	T	0.828	0.172		0.038	0.006	4.30E-09	0.0004	1.000
rs12507026	4	45179317	A	T	0.361	0.361		0.045	0.005	9.55E-19	0.0009	0.000
rs12507026	4	45120599	G	A	0.545	0.455		0.030	0.005	7.72E-10	0.0004	0.998
rs2307111	5	75707853	T	C	0.547	0.453		-0.032	0.005	7.43E-10	0.0004	0.000
rs2307111	5	75684651	C	A	0.231	0.231		-0.017	0.006	0.003681	0.0001	1.000
rs2307111	5	75682020	T	C	0.769	0.231		0.017	0.006	0.003474	0.0001	1.000
rs2206277	6	50830813	C	T	0.190	0.190		0.054	0.006	2.05E-18	0.0009	0.000
rs2206277	6	50917986	G	T	0.566	0.434		0.012	0.005	0.017069	0.0001	1.000
rs2206277	6	50922483	G	A	0.434	0.434		-0.012	0.005	0.015122	0.0001	0.901
rs830463	8	76015930	G	A	0.727	0.273		0.028	0.006	4.86E-07	0.0003	1.000
rs830463	8	76068626	A	G	0.470	0.470		0.030	0.005	6.58E-10	0.0004	0.000
rs830463	8	76010081	T	G	0.274	0.274		-0.028	0.006	5.67E-07	0.0003	1.000
rs3838785	11	27685939	A	C	0.803	0.197		0.032	0.006	3.16E-07	0.0003	0.720
rs3838785	11	27685485	G	A	0.194	0.194		-0.032	0.006	6.10E-07	0.0003	1.000
rs3838785	11	27657463	GT	G	0.579	0.421		-0.030	0.005	3.14E-09	0.0004	0.000
rs7138803	12	49853685	G	A	0.297	0.297		0.036	0.005	1.69E-11	0.0005	0.390
rs9568868	13	53533448	G	T	0.139	0.139		0.047	0.007	5.74E-11	0.0005	0.372
rs1421085	16	53767042	T	C	0.295	0.295		0.090	0.006	6.11E-59	0.0030	0.559
rs1421085	16	53772233	T	C	0.795	0.205		0.061	0.006	1.43E-22	0.0011	0.732
rs6567160	18	60066869	G	A	0.313	0.313		0.033	0.005	6.07E-10	0.0004	1.000
rs6567160	18	60065457	A	G	0.688	0.312		-0.033	0.005	6.86E-10	0.0004	1.000
rs6567160	18	60161902	T	C	0.210	0.210		0.053	0.006	8.22E-19	0.0009	0.000
rs28590228	19	47077985	C	T	0.504	0.496		0.033	0.005	4.76E-10	0.0004	0.299
rs28590228	19	47087297	C	A	0.307	0.307		-0.018	0.006	0.001309	0.0001	0.999
rs111490516	22	29919747	T	C	0.643	0.357		0.009	0.005	0.087543	0.0000	1.000
rs111490516	22	29906934	C	T	0.037	0.037		0.078	0.013	4.52E-09	0.0004	0.000
rs111490516	22	29918103	T	G	0.358	0.358		-0.009	0.005	0.086456	0.0000	1.000
rs1379871	X	31836665	G	C	0.413	0.413		0.029	0.004	1.35E-11	0.0005	1.000
rs1379871	X	31860250	T	C	0.432	0.432		-0.014	0.004	0.000917	0.0001	0.565

Given the great caution with which these results would need to be considered, we have chosen not to include them in our main findings. However, we have clarified in the text why we only present findings assuming one causal variant.

Thus, we have edited the Methods for fine-mapping as follows: “We restricted this analysis to variants located within +/- 100 kb of the locus index variants. We calculated LD using our analysis subset of the TOPMed data. As PAINTOR may be sensitive to misspecification of the number of causal variants, we assumed one causal variant per locus, unless evidence of independent secondary signals within the 100 kb window was identified following conditional analysis, in which case we allowed for additional causal variants per locus. Restriction to a 100 kb +/- window was done due the computational burden when

applied to WGS data and, for most loci, as LD decays at >25 kb.”

And have added the following statement to the results: “Two of our loci identified potential secondary signals following conditional analysis, the *TMEM18/ALKAL2* and *MC4R* loci, but both secondary signals were >100 kb from our index variant.”

Response to reviewer comments (Round 2): Whole Genome Sequencing Analysis of Body Mass Index identifies Novel African Ancestry-Specific Risk Allele

We would like to thank all of the reviewers again for their consideration of our edits and responses to their critiques. We were happy to see that the reviewers largely felt that we addressed their concerns. We have attached a point-by-point response to the reviewers' remaining concerns and recommendations (see below). Our responses are shown in blue, with edits to the paper beyond the last version noted in the response and highlighted in red in the revised manuscript. We again thank you for your time and thoughtful consideration and we hope you find our manuscript improved and all of your concerns addressed.

REVIEWER COMMENTS

Reviewer #1 (Remarks to the Author):

The authors have addressed my concerns.

We thank the reviewer and are happy they are satisfied with our modifications in response to their previous concerns.

Reviewer #2 (Remarks to the Author):

Thank you for addressing many of my concerns. I would still like to see the African-only analysis done with African-specific PCs. Given the higher mean BMI in African ancestry in this study and the higher allele frequency in African ancestry of the novel implicated variant, the best way to make sure that there is no confounding by percent or type of African ancestry in the results is to use African-specific PCs as covariates. It is also field standard to use ancestry-specific PCs when doing an ancestry-specific analysis.

We thank the reviewer for reviewing our previous edits. To address their remaining concern that there may be residual confounding due to population stratification, we have performed the following analysis. We have subset our AFR population and re-calculated subset-specific PCs and re-run the association analysis for our AFR-specific GWAS to evaluate the one novel locus, but also to ensure that there were no differences genome-wide. These results used 10PCs calculated in the same individuals as used in the regression. We used the same covariates as before but replaced the 11 trans-ancestry PCs with 10 AFR-specific PCs. As one would expect when changing any covariates, the p-values are slightly different, but the same SNPs are the index SNPs at all 3 loci and they are all less than than the previously defined genome-level threshold ($P < 5E-9$) and these are still the only 3 significant loci (**See Review Round 2 - Table 1**). Additionally, there was no substantial change in the effect estimate ($ABS(\text{Beta}) > 90\%$ of original Beta). Given that these analyses have not changed the findings or interpretation, and it is not the standard practice within TOPMed to re-calculate PCs in population subsets, we have chosen not to include these results in the manuscript.

Review Round 2 - Table 1. Summary of top loci from sensitivity analysis using subset-specific PCs.

Study	CHR	POS	N	Allele Frequency	Beta	Standard Error	P-value
-------	-----	-----	---	------------------	------	----------------	---------

Sensitivity	1	177920345	22,488	25.04%	0.0741	0.0111	2.15E-11
Main	1	177920345	22,488	25.04%	0.0731	0.0111	4.00E-11
Sensitivity	16	53794050	22,488	10.63%	0.0939	0.0160	3.92E-09
Main	16	53794050	22,488	10.63%	0.0982	0.0159	6.72E-10
Sensitivity	22	29906123	22,488	12.56%	0.0859	0.0145	2.96E-09
Main	22	29906123	22,488	12.56%	0.0873	0.0145	1.66E-09

Reviewer #3 (Remarks to the Author):

Please find my responses to each of the numbered rebuttals. (Note, we removed previous comments that did not require an additional response for this reviewer.)

Major comments

Reviewer – Round 2 (1): Thank you for looking into this, but it seems odd to claim that it is not within scope since your title implies that it is through sequencing that you detect the variant and would not have discovered it otherwise.

We agree with the reviewer that addressing the disparity between imputation quality and sequencing that leads to either new discoveries, non-replication of previous discoveries, or changes in power is an interesting topic. However, our focus for this paper was to leverage the sequencing for discovery of new variants, understanding that there are multiple reasons as to why new discoveries may be possible. Through whole genome sequencing of a large number of ancestrally diverse individuals, a much larger number of variants are able to be imputed to the truth. However, data use agreements for some studies within TOPMed provide limitations on the use of data, including for imputation. Thus, it would breach our agreements to use these data for those purposes. Further, the largest discovery analysis of BMI to date is heavily biased towards European ancestry individuals and is only imputed to HapMap3 and HRC. Teasing apart the cause of the novel discovery is a very broad topic that could be covered in a separate paper. Indeed, there are publications that are looking at the improvement in imputation with the advent of more diverse reference panels. <https://www.biorxiv.org/content/10.1101/2023.05.22.541241v1.full>.

Reviewer – Round 2 (2): Thank you for providing information to aid in reproducibility of results. However, the link seems to only include the quality control of datasets and variant calling, but not the analyses presented in this paper. Zenodo links to the code used to perform your analyses would go a long way to ensuring reproducibility of the presented results.

Response – Round 2: We apologize for the confusion. We understand that there are many links presented. The code for our analysis was carried out using this specific pipeline: https://github.com/AnalysisCommons/genesis_wdl, under the R scripts “genesis_nullmodel.R”, “genesis_tests.R”, “pipelineFunctions.R”, etc.

Minor comments

Reviewer – Round 2 (2): Excellent, thank you. Further clarification described in the reviewer response for why LOFTEE and other tools were chosen would be useful to include in article methods section.

Response – Round 2:

Thank you for the recommendation. We have now edited our previous response under ‘Rare Variant Aggregate Association Analysis’ section, as shown below.

Rare Variant Aggregate Association Analysis

Rare variants with a MAF $\leq 1\%$ were tested in aggregate by gene unit across studies in the multi-population analysis. Variants were grouped into gene units in reference to GENCODE v28, including both coding variants and variants falling within gene-associated non-coding elements. Coding variants included high-confidence loss of function variants **annotated by LOFTEE** (Ensembl VEP LoF = HC), missense variants (MetaSVM score > 0) and in-frame insertion/deletions or synonymous variants (FATHMM-XF coding score > 0.5). In addition to coding variants, we included variants falling within the promoter of each transcript tested. Promoter regions were defined as falling in the 5 kb region 5’ of the transcript and also overlaying a FANTOM5 Cage Peak. In order to identify regulatory elements likely to be acting through the tested gene, we leveraged the GeneHancer database. GeneHancer identifies enhancer regions and associates them with the specific genes they are likely to regulate, allowing us to aggregate regulatory regions by the likely target gene. GeneHancer regions were limited to the top 50% scored regions and variants falling in these regulatory elements were further filtered to those most likely to have a functional impact (FATHMM-MKL noncoding score > 0.75). Variants aggregated to gene units were tested using variant-set mixed model association tests (SMMAT). Variants were weighted inversely to their MAF using a beta distribution density function with parameters 1 and 25. Genes were considered significantly associated after Bonferroni correction for the number of genes analyzed ($P < 5 \times 10^{-7}$). **These annotations were selected by the TOPMed Data Coordinating Center (DCC) as part of the centralized and harmonized annotations, and were chosen to focus on annotation that was previously shown to have high agreement with other annotation resources and high prediction accuracy.**

Reviewer – Round 2 (3): Thank you. It would be good to gain an understanding for why the remaining hits in Akbari were not found - is it simply a power issue? You mention the increased flexibility of GENESIS allowing for heterogeneity of effect size. It would be helpful to reference this advantage in the methods, and why other approaches were not used. The motivation and backing up of your methods over the meta-analysis as suggested by reviewers 1 and 2 is important and should be discussed in the article (and not relegated to the reviewer rebuttal).

Response – Round 2:

Replication of gene-based findings is extremely nuanced. Akbari et al. had a much larger samples (~7 times larger), including multiple studies with related individuals. In addition to sample size, differences in alleles harbored by individuals in a given dataset, grouping of variants, and rare variant test all can have large impact on significance of gene-based test. Failure to replicate gene-based signals must be interpreted with extreme caution. In addition, the results published by Akbari et al. presented findings at multiple genes, but they only attempted to replicate and validate one gene-based association, therefore there is still question about the robustness of the remaining findings and whether we should expect to be able to replicate them. Therefore, it is likely a combination of power due to differences in sample size, underlying variant selection, and difference in methods for gene-based analysis. We have added further

explanation about the possible lack of replication in the Discussion section. Also, we have now provided further clarification on GENESIS for our analysis plan:

“Common Variant Association Analysis

We performed multi-population WGS association analysis of BMI using GENESIS on the Analysis Commons (<http://analysiscommons.com>) computation platform. GENESIS was chosen due to its analytical flexibility in relationship to allowing for heterogeneity of effect by population group, an option well-suited to the demographic and genetic background of our study population. Analyses were performed using linear mixed models (LMM). To improve power and control for false positives with a non-normal phenotype distribution, we implemented a fully adjusted two-stage procedure for rank-normalization when fitting the null model¹⁵...

...Due to the large number of variants tested (N = 90,142,062) in the multi-population analysis, we adopted a significance threshold of 5×10^{-9} as has been used previously¹⁹. This approach maximizes participant inclusion, thereby maximizing the sample size in our discovery cohort to increase statistical power and avoiding the misinterpretation of group-specific effects in underpowered strata. By including ancestrally diverse populations, we leverage differences in LD across populations, which has been shown to help identify novel loci, narrow down causal variants, and improve the variance explained in models (PMIDs: 26748518, 31217584, 36119389). For quantitative traits like BMI, multi-population analyses that account for heteroscedasticity of genetic effects across population groups have proven effective in increasing study power and reducing genomic inflation (PMIDs: 26748518, 31217584). Lastly, and most importantly, a pooled approach aids in decreasing health disparities by identifying loci that generalize across populations.

Additionally, group-specific analyses were conducted in the two largest population groups, European and African, to determine whether a particular group with large sample size is driving the observed association signal.

“In addition to our novel findings, 17 of the 18 identified variants reside in previously reported BMI-associated loci, highlighting the generalizability of the genes underlying BMI across populations, including *SEC16B*, *TMEM18*, *ETV5*, *GNPDA2*, *BDFN*, and *MC4R*. Three of the loci harbor genes implicated in severe and early-onset obesity – *ADCY3*, *BDNF*, and *MC4R*. We also consistently identified multiple association signals of high effect in *MC4R*, which is a well-established monogenic obesity gene, through our discovery analysis, internal conditional analysis, and rare variant aggregate analysis. Despite not identifying novel SNPs in *MC4R* that are independent of known BMI-associated SNPs, we replicated a secondary signal in this gene, rs2229616, a rare missense variant previously reported in individuals of European ancestry by Speliotes et al. In addition to *MC4R*, three other genes – *ROBO1*, *GPR151*, and *ANO4* – of the exome-wide significant genes identified in Akbari et al. showed nominal significance in our rare variant aggregate analysis. This lends further support to the generalizability of these genes across populations, given that 85% were of European ancestry and 15% of admixed American ancestry in Akbari et al., compared to our more diverse cohort with 49% European and 51% other populations. We did not replicate 12 of the gene-based findings from Akbari et al. The previous study had nearly seven-fold larger sample size compared to the current study, included related individuals increasing the opportunity for multiple copies of rare variants, and implemented alternative methods for gene-based analysis and variant binning. Therefore, it is likely a combination of power due to differences in sample size, underlying variant selection, difference in methods for gene-based analysis, and potential winner’s curse that contributed to the lack of validation of their findings in the current study.”

Reviewer – Round 2 (8) Reviewer – As PAINTOR may be sensitive to misspecification of the number of causal variants' seems an understatement. Thank you to the authors for investigating. Your examination of the loci here is helpful and should be included in the supplement of the paper.

What is your reference for >25kb LD decays, and what does this mean? $R^2 < X$ for some large proportion Y% of the time? What are X and Y? Without X and Y, the statement doesn't tell us a great deal. What is the LD between your primary signal and secondary signals? Doesn't the secondary signal being >25kb away from your primary signal contradict your LD decay claim?

We based our window size on the recommendations of Greenbaum and Deng 2017 (PMID: 28425624) that suggest a window of 100 kb based on the LD decay >25 kb. The observation of decay >25 kb is based on evidence of the portability of LD blocks up to 20 kb windows across multiple HapMap populations (>80% of haplotype blocks less than this distance) with a decrease in portability when extending this window, among other analyses that point to decreased LD decay ($R^2 < 0.5$) at greater distances identified by Conrad DF, et al. 2006 (PMID: 17057719). Given that the secondary signal is greater than 25 kb from our primary signal, this supports the conclusion that these are independent loci, given that we expect increased independence (lower LD) at >25 kb distances. However, we do acknowledge that limiting the window size can cause us to potentially miss causal variants that may be greater than 100 kb from our lead variant. We have modified our fine-mapping methods section to provide a reference for the window chosen and acknowledge this limitation.

We appreciate that the reviewer found our additional PAINTOR analyses helpful; however, given that we are not confident in the results of PAINTOR assuming more than one causal variant for the reasons previously noted, we have chosen to leave these analyses out of the main paper and reserve this to the response to reviewers that will be available alongside the paper. We have edited the methods to clarify further why only one causal variant was assumed:

“We restricted this analysis to variants located within ± 100 kb of the locus index variants. We calculated LD using our analysis subset of the TOPMed data. As PAINTOR may be sensitive to the misspecification of the number of causal variants, **limiting our ability to interpret findings in the absence of evidence for more than one signal**, we assumed one causal variant per locus, unless evidence of independent secondary signals within the 100 kb window was identified following conditional analysis, in which case we allowed for additional causal variants per locus. Restriction to a 100 kb \pm window was applied due to the computational burden of WGS data and **because**, for most loci, LD decays at > 25 kb (PMIDs: 28425624, 17057719). **While we extend our fine-mapping locus to 100 kb to allow for potential LD beyond 25 kb, one limitation of this approach is that we may still miss potentially causal variants that are >100 kb from our index SNP, including secondary signals.**”

Response to reviewer comments (Round 3): Whole Genome Sequencing Analysis of Body Mass Index identifies Novel African Ancestry-Specific Risk Allele

We would like to thank the reviewers again for their consideration of our edits and responses to their critiques. We were happy to see that the reviewers largely felt that we addressed their concerns. To address the remaining comments and suggestions, we have included a point-by-point response below. Our responses are presented in blue, any further edits to the manuscript since the last version are indicated in the responses and highlighted in red within the revised manuscript, and previous comments still included are greyed out. We greatly appreciate your time and consideration and hope you find the manuscript improved and all concerns satisfactorily resolved.

Reviewer #2 (Remarks to the Author):

the authors have addressed my concerns

We thank the reviewer and are happy they are satisfied with our modifications in response to their previous concerns.

Reviewer #3 (Remarks to the Author):

Firstly, regarding the response to reviewer 2, I really don't understand why the results you present (in the table) cannot be included in the manuscript. They back up your use of mega-analyses, which as the reviewer mentions, is not standard practice. The statement "it is not the standard practice within TOPMed to re-calculate PCs in population subsets", seems an odd reason not to include the additional results...

We have now included the results table from our previous response to reviewer 2 into the supplement (new Supplementary Data 7), and have edited the methods and results sections as follows:

Methods:

“Additionally, group-specific analyses were conducted in the two largest population groups, European and African, to determine whether a particular group with large sample size is driving the observed association signals. **To address any concerns over potential residual confounding due to population substructure in our African group-specific GWAS, especially for our novel association that differed in allele frequency across population groups, we conducted a sensitivity analysis using group-specific PCs. The sensitivity analysis used 10 African group-specific PCs calculated in the same individuals as used in the pooled analysis and the same covariates as before.**”

Results:

“Notably, the novel locus in *MTMR3* achieved significance exclusively in the African group. While the most significant SNP in the African population group (rs73396827) differed from that in the multi-population analysis (rs111490516), the two were in strong LD in the TOPMed African population ($R^2 = 1.00$). Both of these SNPs were fixed in the European group (MAF = 0%). **Our sensitivity analysis illustrates the robustness of our findings as we identified the same three significant association signals ($P < 5 \times 10^{-9}$) and lead SNPs as in the main analysis (Supplementary Data 7). Additionally, there was no substantial change in the effect estimate (directionally consistent and sensitivity $|\text{Beta}| > 90\%$ original $|\text{Beta}|$).**”

...In fact, I would encourage the sharing of the entire AFR only GWAS sumstats alongside your primary mega meta-analysis sumstats.

We are sorry if this was unclear before, but our intention is to share all of our GWAS summary statistics publicly that were used in this paper, which would include the pooled-population, African, and European GWAS summary statistics. We have made this clear in our “Data and Code Availability” section as follows:

“All summary statistics from our GWAS analyses presented, **including pooled, African, European, and sensitivity**, are provided through the NHGRI-EBI Catalog of human genome-wide association studies (<https://www.ebi.ac.uk/gwas/>).

Reviewer #3 (Remarks to the Author): Please find my responses to each of the numbered rebuttals. (Note, we removed previous comments that did not require an additional response for this reviewer.)

Major comments

Reviewer – Round 2 (1): Thank you for looking into this, but it seems odd to claim that it is not within scope since your title implies that it is through sequencing that you detect the variant and would not have discovered it otherwise.

We agree with the reviewer that addressing the disparity between imputation quality and sequencing that leads to either new discoveries, non-replication of previous discoveries, or changes in power is an interesting topic. However, our focus for this paper was to leverage the sequencing for discovery of new variants, understanding that there are multiple reasons as to why new discoveries may be possible. Through whole genome sequencing of a large number of ancestrally diverse individuals, a much larger number of variants are able to be imputed to the truth. However, data use agreements for some studies within TOPMed provide limitations on the use of data, including for imputation. Thus, it would breach our agreements to use these data for those purposes. Further, the largest discovery analysis of BMI to date is heavily biased towards

European ancestry individuals and is only imputed to HapMap3 and HRC. Teasing apart the cause of the novel discovery is a very broad topic that could be covered in a separate paper. Indeed, there are publications that are looking at the improvement in imputation with the advent of more diverse reference panels.

<https://www.biorxiv.org/content/10.1101/2023.05.22.541241v1.full>.

Thank you. Perhaps mentioning evidence of poorer imputation accuracy in non-European populations is likely to be an issue, even for e.g. the 'global' screening array, means that it makes sense to look in WGS data, which is becoming more and more ubiquitous, acknowledging that you're not actually able to test this assertion in your data - highlighting the data use agreements, in the discussion.

We thank the reviewer for this suggestion. We have now edited the discussion as follow:

“Therefore, the lack of discovery in prior publications may not be due to insufficient power. As indicated by our fine-mapping results and potential regulatory role of nearby variants in this novel locus, our index SNP is likely not causal but could be in LD with a causal SNP and also poorly captured in studies relying on imputation. In other words, the causal variant underlying this locus may be nearby, less frequent, and on an LD block more frequent in a population poorly represented in other imputation reference panels, but well represented in our WGS and highly diverse sample (e.g., Caribbean admixed individuals). **The non-European ancestry populations particularly fall short in imputation performance due to their persistent underrepresentation in reference panels and overestimate of imputation accuracy (PMID: 38604166).** In this case, one would require sequencing data in a large sample size with the relevant haplotype to detect a significant association that was not able to be identified with imputation in a similar number of people. **Unfortunately, we are unable to test this hypothesis in our data due to data access constraints.** Thus, future studies with WGS in study populations with genetic similarity and functional follow-work are needed to further narrow in on the causal variant(s) underlying this association signal.”

Reviewer – Round 2 (2): Thank you for providing information to aid in reproducibility of results. However, the link seems to only include the quality control of datasets and variant calling, but not the analyses presented in this paper. Zenodo links to the code used to perform your analyses would go a long way to ensuring reproducibility of the presented results.

Response – Round 2: We apologize for the confusion. We understand that there are many links presented. The code for our analysis was carried out using this specific pipeline:

https://github.com/AnalysisCommons/genesis_wdl, under the R scripts “genesis_nullmodel.R”, “genesis_tests.R”, “pipelineFunctions.R”, etc.

Thank you. This makes sense, but https://github.com/AnalysisCommons/genesis_wdl, is general use code for pipelining GENESIS, rather than your code to perform call GENESIS in your primary analysis, and all of the subsequent code that you would have used for your subsequent post-GWAS analysis. Please include your code for GWAS (including all options passed in the wdl), and post-GWAS analysis (including all versions of software used, and options/flags used therein, and all scripts), with Zenodo.

Thank you for these comments. We have created a github repo (<https://github.com/Justice-Genetics-Lab/TOPMed-WGS-BMI-GWAS/tree/main>) including all scripts used in this manuscript with referenced software packages along with the parameters used as noted in the publication. These include bash scripts used to launch analyses on the DNAnexus platform. We have added the following statement to the Data and Code Availability section:

“Code and scripts used in study analyses: All scripts used for running analyses on the TOPMed Analysis Commons on the DNAnexus Platform are provided here: <https://github.com/Justice-Genetics-Lab/TOPMed-WGS-BMI-GWAS/tree/main>.”

Minor comments Reviewer – Round 2 (2): Excellent, thank you. Further clarification described in the reviewer response for why LOFTEE and other tools were chosen would be useful to include in article methods section.

Response – Round 2: Thank you for the recommendation. We have now edited our previous response under ‘Rare Variant Aggregate Association Analysis’ section, as shown below.

Rare Variant Aggregate Association Analysis

Rare variants with a $MAF \leq 1\%$ were tested in aggregate by gene unit across studies in the multi population analysis. Variants were grouped into gene units in reference to GENCODE v28, including both coding variants and variants falling within gene-associated non-coding elements. Coding variants included high-confidence loss of function variants annotated by LOFTEE (Ensembl VEP LoF = HC), missense variants (MetaSVM score > 0) and in-frame insertion/deletions or synonymous variants (FATHNMM-XF coding score > 0.5). In addition to coding variants, we included variants falling within the promoter of each transcript tested. Promoter regions were defined as falling in the 5 kb region 5’ of the transcript and also overlaying a FANTOM5 Cage Peak. In order to identify regulatory elements likely to be acting through the tested gene, we leveraged the GeneHancer database. GeneHancer identifies enhancer regions and associates them with the specific genes they are likely to regulate, allowing us to aggregate regulatory regions by the likely target gene. GeneHancer regions were limited to the top 50% scored regions and variants falling in these regulatory elements were further filtered to those most likely to have a functional impact (FATHMM-MKL noncoding score > 0.75).

Variants aggregated to gene units were tested using variant-set mixed model association tests (SMMAT). Variants were weighted inversely to their MAF using a beta distribution density function with parameters 1 and 25. Genes were considered significantly associated after Bonferroni correction for the number of genes analyzed ($P < 5 \times 10^{-7}$). These annotations were selected by the TOPMed Data Coordinating Center (DCC) as part of the centralized and harmonized annotations, and were chosen to focus on annotation that was previously shown to have high agreement with other annotation resources and high prediction accuracy.

Thank you. Can you provide some references for your assertion that “chosen to focus on annotation that was previously shown to have high agreement with other annotation resources and high prediction accuracy.”

We apologize if this was unclear in our response letter. While we did not include the associated reference in the response letter, it was cited in the manuscript as reference 60:

Liu, X., Wu, C., Li, C. & Boerwinkle, E. dbNSFP v3.0: A One-Stop Database of Functional Predictions and Annotations for Human Nonsynonymous and Splice-Site SNVs. *Hum Mutat* 37,831 235-41 (2016).

Reviewer – Round 2 (3): Thank you. It would be good to gain an understanding for why the remaining hits in Akbari were not found - is it simply a power issue? You mention the increased flexibility of GENESIS allowing for heterogeneity of effect size. It would be helpful to reference this advantage in the methods, and why other approaches were not used. The motivation and backing up of your methods over the meta-analysis as suggested by reviewers 1 and 2 is important and should be discussed in the article (and not relegated to the reviewer rebuttal).

Response – Round 2: Replication of gene-based findings is extremely nuanced. Akbari et al. had a much larger samples (~7 times larger), including multiple studies with related individuals. In addition to sample size, differences in alleles harbored by individuals in a given dataset, grouping of variants, and rare variant test all can have large impact on significance of gene-based test. Failure to replicate gene-based signals must be interpreted with extreme caution. In addition, the results published by Akbari et al. presented findings at multiple genes, but they only attempted to replicate and validate one gene-based association, therefore there is still question about the robustness of the remaining findings and whether we should expect to be able to replicate them. Therefore, it is likely a combination of power due to differences in sample size, underlying variant selection, and difference in methods for gene-based analysis. We have added further explanation about the possible lack of replication in the Discussion section. Also, we have now provided further clarification on GENESIS for our analysis plan:

“Common Variant Association Analysis We performed multi-population WGS association analysis of BMI using GENESIS on the Analysis Commons (<http://analysiscommons.com>)

computation platform. GENESIS was chosen due to its analytical flexibility in relationship to allowing for heterogeneity of effect by population group, an option well-suited to the demographic and genetic background of our study population. Analyses were performed using linear mixed models (LMM). To improve power and control for false positives with a non-normal phenotype distribution, we implemented a fully adjusted two-stage procedure for rank normalization when fitting the null model¹⁵...

...Due to the large number of variants tested ($N = 90,142,062$) in the multi-population analysis, we adopted a significance threshold of 5×10^{-9} as has been used previously¹⁹. This approach maximizes participant inclusion, thereby maximizing the sample size in our discovery cohort to increase statistical power and avoiding the misinterpretation of group-specific effects in underpowered strata. By including ancestrally diverse populations, we leverage differences in LD across populations, which has been shown to help identify novel loci, narrow down causal variants, and improve the variance explained in models (PMIDs: 26748518, 31217584, 36119389). For quantitative traits like BMI, multi-population analyses that account for heteroscedasticity of genetic effects across population groups have proven effective in increasing study power and reducing genomic inflation (PMIDs: 26748518, 31217584). Lastly, and most importantly, a pooled approach aids in decreasing health disparities by identifying loci that generalize across populations.

Additionally, group-specific analyses were conducted in the two largest population groups, European and African, to determine whether a particular group with large sample size is driving the observed association signal.

“In addition to our novel findings, 17 of the 18 identified variants reside in previously reported BMI associated loci, highlighting the generalizability of the genes underlying BMI across populations, including SEC16B, TMEM18, ETV5, GNPDA2, BDFN, and MC4R. Three of the loci harbor genes implicated in severe and early-onset obesity – ADCY3, BDNF, and MC4R. We also consistently identified multiple association signals of high effect in MC4R, which is a well-established monogenic obesity gene, through our discovery analysis, internal conditional analysis, and rare variant aggregate analysis. Despite not identifying novel SNPs in MC4R that are independent of known BMI-associated SNPs, we replicated a secondary signal in this gene, rs2229616, a rare missense variant previously reported in individuals of European ancestry by Speliotes et al. In addition to MC4R, three other genes – ROBO1, GPR151, and ANO4 – of the exome-wide significant genes identified in Akbari et al. showed nominal significance in our rare variant aggregate analysis. This lends further support to the generalizability of these genes across populations, given that 85% were of European ancestry and 15% of admixed American ancestry in Akbari et al., compared to our more diverse cohort with 49% European and 51% other populations. We did not replicate 12 of the gene-based findings from Akbari et al. The previous study had nearly sevenfold larger sample size compared to the current study, included related

individuals increasing the opportunity for multiple copies of rare variants, and implemented alternative methods for gene-based analysis and variant binning. Therefore, it is likely a combination of power due to differences in sample size, underlying variant selection, difference in methods for gene-based analysis, and potential winner's curse that contributed to the lack of validation of their findings in the current study.”

Thank you.

Reviewer – Round 2 (8) Reviewer – As PAINTOR may be sensitive to misspecification of the number of causal variants' seems an understatement. Thank you to the authors for investigating. Your examination of the loci here is helpful and should be included in the supplement of the paper.

What is your reference for $>25\text{kb}$ LD decays, and what does this mean? $R^2 < X$ for some large proportion $Y\%$ of the time? What are X and Y ? Without X and Y , the statement doesn't tell us a great deal. What is the LD between your primary signal and secondary signals? Doesn't the secondary signal being $>25\text{kb}$ away from your primary signal contradict your LD decay claim?

We based our window size on the recommendations of Greenbaum and Deng 2017 (PMID: 28425624) that suggest a window of 100 kb based on the LD decay $>25\text{ kb}$. The observation of decay $>25\text{ kb}$ is based on evidence of the portability of LD blocks up to 20 kb windows across multiple HapMap populations ($>80\%$ of haplotype blocks less than this distance) with a decrease in portability when extending this window, among other analyses that point to decreased LD decay ($R^2 < 0.5$) at greater distances identified by Conrad DF, et al. 2006 (PMID: 17057719). Given that the secondary signal is greater than 25 kb from our primary signal, this supports the conclusion that these are independent loci, given that we expect increased independence (lower LD) at $>25\text{ kb}$ distances. However, we do acknowledge that limiting the window size can cause us to potentially miss causal variants that may be greater than 100 kb from our lead variant. We have modified our fine-mapping methods section to provide a reference for the window chosen and acknowledge this limitation.

We appreciate that the reviewer found our additional PAINTOR analyses helpful; however, given that we are not confident in the results of PAINTOR assuming more than one causal variant for the reasons previously noted, we have chosen to leave these analyses out of the main paper and reserve this to the response to reviewers that will be available alongside the paper. We have edited the methods to clarify further why only one causal variant was assumed:

“We restricted this analysis to variants located within $\pm 100\text{ kb}$ of the locus index variants. We calculated LD using our analysis subset of the TOPMed data. As PAINTOR may be sensitive to the misspecification of the number of causal variants, limiting our ability to interpret findings in

the absence of evidence for more than one signal, we assumed one causal variant per locus, unless evidence of independent secondary signals within the 100 kb window was identified following conditional analysis, in which case we allowed for additional causal variants per locus. Restriction to a 100 kb \pm window was applied due the computational burden of WGS data and because, for most loci, LD decays at > 25 kb (PMIDs: 28425624, 17057719). While we extend our fine-mapping locus to 100 kb to allow for potential LD beyond 25 kb, one limitation of this approach is that we may still miss potentially causal variants that are >100 kb from our index SNP, including secondary signals.”

I disagree with the exclusion of the comparative PAINITOR analysis assuming multiple causal variants. It is useful given the statement in PMID: 26189819 “In general, we find that trans-ethnic fine-mapping strategies that assume a single causal variant are less optimal than those that allow for multiple causal variants”, from the PAINITOR authors. Furthermore, reference 64 seems to suggest that PAINITOR can often return unreasonable results even for a single causal variant, which would seem to suggest that other finemapping approaches should be considered for consistency. Also, both advocate for the use of GWAS within population specific LD, and cross-ancestry finemapping. Stating this in the discussion to improve finemapping would be advisable.

We thank the reviewer for their insightful comments. We agree that biologically speaking allowing multiple causal variants/haplotypes is ideal. However, our experience, both in this work and in other work, has shown that relaxing the assumption of a single signal leads to spurious and inconsistent results across approaches. These observations have been echoed in others work as well. Ultimately, because we do not have any secondary signals within 100kb, we do not expect there to be more than one causal variant/haplotype. To further illustrate this to potential readers, we have edited the Discussion as follows:

“Beyond these two high confident SNPs, five additional SNPs in as many loci were identified in our fine-mapping analysis with a PP > 0.5 , all with moderate to high evidence of a regulatory or functional role related to a nearby gene. Given that one of these SNPs, rs1421085 in *FTO*, has been successfully confirmed as a causal variant at this locus, additional variants highlighted in our fine-mapping analysis warrant consideration in future functional studies. **While the use of population-matched LD is ideal for trans-population fine-mapping, such data remain scarce in non-European populations (PMID: 26157023). Although trans-population fine-mapping strategies assuming multiple causal variants are considered superior (PMID: 26189819), PAINITOR may yield unreliable results under certain conditions even when assuming a single variant (PMID: 39187616). Even though we are encouraged that Kanai et al., using FINEMAP (PMID: 26773131) and SuSiE (PMID: 37220626), identified the same two variants as high-confidence causal variants, cross-validation of results using other fine-mapping approaches remains necessary.**”